# Towards the Efficient Inference by Incorporating Automated Computational Phenotypes under Covariate Shift

Chao Ying [1 2]  Jun Jin [3]  Yi Guo [4]  Xiudi Li [5]  Muxuan Liang [6]  Jiwei Zhao [1 2]

## Abstract

Collecting gold-standard phenotype data via manual extraction is typically labor-intensive and slow, whereas automated computational phenotypes (ACPs) offer a systematic and much faster alternative. However, simply replacing the gold-standard with ACPs, without acknowledging their differences, could lead to biased results and misleading conclusions. Motivated by the complexity of incorporating ACPs while maintaining the validity of downstream analyses, in this paper, we consider a semi-supervised learning setting that consists of both labeled data (with gold-standard) and unlabeled data (without gold-standard), under the covariate shift framework. We develop doubly robust and semiparametrically efficient estimators that leverage ACPs for general target parameters in the unlabeled and combined populations. In addition, we carefully analyze the efficiency gains achieved by incorporating ACPs, comparing scenarios with and without their inclusion. Notably, we identify that ACPs for the unlabeled data, instead of for the labeled data, drive the enhanced efficiency gains. To validate our theoretical findings, we conduct comprehensive synthetic experiments and apply our method to multiple real-world datasets, confirming the practical advantages of our approach. Code: 

## 1. Introduction

Automated computational phenotype (ACP) refers to the process of using advanced algorithms and models, including but not limited to, machine learning, natural language processing (NLP), large language model (LLM) and generative AI, to automatically compute and define phenotypes from complex data such as electronic health records (EHRs). Though obtaining gold-standard phenotype data via manual extraction is labor-intensive and slow, ACP allows researchers to obtain phenotypes systemically and quickly, scaling their efforts in precision medicine and supporting the broader goal of turning raw data into actionable insights for research and healthcare improvements.

Despite these benefits, simply replacing gold-standard data with ACPs in downstream analyses introduces new challenges. As computational phenotyping algorithms are often trained to minimize prediction error such as the mean squared error (MSE), ACPs may contain a non-negligible bias for important downstream tasks such as statistical inference. Therefore, the indiscriminate use of ACPs, without acknowledging their distinction from gold-standard phenotype data, can lead to biased results and misleading conclusions. In this work, instead of replacing gold-standard phenotype data, we investigate the benefits of appropriately incorporating these ACPs, in order to enhance the estimation efficiency and to achieve the more powerful statistical inference.

In many applications, gold-standard phenotypes (labels) are only available for a small subset of individuals due to the high cost of manual extraction, while unlabeled data are more readily accessible. Meanwhile, the selection of labeled individuals is often based on baseline characteristics rather than being purely random. This creates a semi-supervised learning setting, where the covariate distribution differs between labeled and unlabeled data. In this paper, we aim to integrate ACPs into classical semi-supervised learning frameworks to enable efficient estimation and inference.

Our motivating study is an analysis on diabetes among children and adolescents from the UF (University of Florida) health system (Li et al., 2025), where diabetes status is adjudicated by medical experts through manual chart reviews for

[1]Department of Statistics, University of Wisconsin-Madison, Madison, Wisconsin, USA [2]Department of Biostatistics & Medical Informatics, University of Wisconsin-Madison, Madison, Wisconsin, USA [3]Henry Ford Health, Detroit, Michigan, USA [4]Department of Health Outcomes & Biomedical Informatics, University of Florida, Gainesville, Florida, USA [5]Biostatistics Division, University of California, Berkeley, California, USA [6]Department of Biostatistics, University of Florida, Gainesville, Florida, USA. Correspondence to: Jiwei Zhao <jiwei.zhao@wisc.edu>.

*Proceedings of the 42$^{nd}$ International Conference on Machine Learning*, Vancouver, Canada. PMLR 267, 2025. Copyright 2025 by the author(s).

a subset of patients selected via stratified random sampling, while ACPs are available for all patients in the EHR through a previously validated decision-tree-based algorithm. The stratified sampling introduces potential covariate shift between the chart-review population and the general EHR population. Our work focuses on how to incorporate these ACPs with chart review data to improve parameter estimation and inference in a prediction model under possible covariate shifts.

## 1.1. Related Work

**Semi-supervised learning and inference**. Semi-supervised learning has been popular in both machine learning and statistics in the past several decades (Zhu, 2005; Chapelle et al., 2009). Based on different assumptions, computational algorithms and estimation methods have been proposed in classification (Rigollet, 2006; Wang et al., 2022), non-parametric regression (Wasserman & Lafferty, 2008), and semi-supervised regression (Kostopoulos et al., 2018), to understand the benefits of the abundant unlabeled data.

In particular, there are considerable progresses in the past several years on how to make use of the unlabeled data for parameter estimation, either with a faster convergence rate or with a smaller asymptotic variance. For example, Chakrabortty & Cai (2018) and Azriel et al. (2022) proposed estimators that are more efficient than the ordinary least square (OLS) that uses labeled data only, under the assumption-lean regression framework (Buja et al., 2019; Berk et al., 2019). Similar conclusions could also be found under the general M-estimation framework (Song et al., 2023). The idea was also extended to the high-dimensional setting where the number of features is allowed to be greater than the sample size (Zhang et al., 2019; Zhang & Bradic, 2022; Cai & Guo, 2020; Deng et al., 2024), as well as in the situation without the sparsity assumption (Livne et al., 2022; Hou et al., 2023).

**Distribution shift**. Distribution shift (Quinonero-Candela et al., 2008) refers to the phenomenon that the joint distributions between the training and testing data are different, or, in our context, between the labeled and unlabeled data. Therefore, the knowledge learnt from the labeled data may no longer be appropriate to be directly used in the unlabeled data or in the combined data. This also motivates the study of unsupervised domain adaptation (Kouw & Loog, 2021), aiming to address the distributional shift between the labeled and unlabeled data domains.

Two dominating types of distribution shifts are covariate shift (the marginal distribution of the feature changes) and label shift (the marginal distribution of the label changes; see, e.g., Storkey (2009); Zhang et al. (2013); Du Plessis & Sugiyama (2014); Iyer et al. (2014); Nguyen et al. (2016); Tasche (2017); Lipton et al. (2018); Garg et al. (2020); Tian et al. (2023); Kim et al. (2024); Lee et al. (2025)). They are suitable assumptions under different contexts. Covariate shift, the focus in this paper, aligns with the causal learning setting (Schölkopf et al., 2012) and has been studied comprehensively in the literature (Shimodaira, 2000; Huang et al., 2006; Sugiyama et al., 2008; Gretton et al., 2009; Sugiyama & Kawanabe, 2012; Kpotufe & Martinet, 2021; Aminian et al., 2022; Rodemann et al., 2023).

**Prediction-powered inference**. To our best knowledge, inference with predicted data derived from black-box AI/ML models started from Wang et al. (2020), followed by Motwani & Witten (2023). More recently, Angelopoulos et al. (2023a) proposed the approach termed *prediction-powered inference* (PPI), which yields valid inference even when the predictions were of a low quality. However, PPI might be worse, in terms of estimation efficiency, than the benchmark method that uses labeled data only. This motivates a growing literature of research that investigates the statistical efficiency gains from different perspectives, such as Angelopoulos et al. (2023b); Miao et al. (2023); Gronsbell et al. (2024); Miao et al. (2024); Ji et al. (2025).

**Surrogacy in biostatistics and causal inference**. In the literature of biostatistics especially clinical trails, there is a line of research on identifying, evaluating and validating surrogate variables, motivated by the fact that primary endpoint may be invasive, costly, or take a long time to measure. This dates back to Prentice (1989) who first proposed the definition and the criterion for surrogacy. Readers of interest could refer to Elliott (2023) for a comprehensive review of surrogate evaluation. This type of research usually seeks the replacement of primary endpoint with surrogate variable, enabling traditional or provisional approval of treatment in clinical trials. Not surprisingly, the replacement process often requires stringent, unverifiable assumptions, such as the strong surrogate assumption in Prentice (1989). More recently, surrogate variables were also analyzed in the causal inference framework to enhance the efficiency of treatment effect estimation on long-term or primary outcomes; see, e.g., Athey et al. (2019); Gupta et al. (2019); Athey et al. (2020); Cheng et al. (2021); Tran et al. (2023); Zhang et al. (2023); Kallus & Mao (2024); Imbens et al. (2024); Gao et al. (2024).

Additionally, surrogate variable is relevant to the misclassification problem or the label noise issue in machine learning (e.g., Lawrence & Schölkopf 2001; Scott 2015; Li et al. 2021; Liu et al. 2023; Guo et al. 2024), also the measurement error issue in statistical literature (e.g., Carroll et al. 2006; Buonaccorsi 2010; Yi 2017; 2021).

## 1.2. Our Contributions

Firstly, we consider a semi-supervised learning setting under the covariate shift framework; that is, the labeling mecha-

nism may depend on features so might not necessarily be purely random. As discussed in our motivating study, the labeling mechanism is often unknown in applications, making it more flexible to assume dependence on features (covariate shift) rather than purely random. In PPI, Angelopoulos et al. (2023a) did consider scenarios involving covariate shift; however, their method assumed that the difference of the two distributions is completely known. Certainly this is not feasible in practice.

Secondly, our assumption on how the ACPs are generated is flexible. The ACP, denoted as $\widehat{Y}$ in the paper, does not have to be an accurate prediction of the ground truth $Y$. Indeed, $\widehat{Y}$ does not have to be in the same magnitude as $Y$, or even the same format as $Y$. For example, $\widehat{Y}$ could be a continuous variable even if $Y$ is binary. In addition, we assume the generating algorithm of $\widehat{Y}$ is somewhat black-box, which, beyond the available feature $\mathbf{X}$ in the labeled and unlabeled data, might also depend on some additional unseen feature, say, $\mathbf{Z}$.

Under this general framework, we demonstrate that the proposed estimator in this paper ensures that incorporating ACPs never reduces estimation efficiency. The only scenario with no efficiency gain occurs when the generation of ACPs depends solely on the available feature $\mathbf{X}$. This aligns with our intuition, simply because in such an extreme situation the ACP $\widehat{Y}$ does not bring in any *new* information. Furthermore, we highlight that the efficiency gain arises specifically from the ACPs in the unlabeled data. In contrast, ACPs in the labeled data do not contribute to the efficiency gain. This is intuitive since the labeled data already contain the ground truth $Y$, so a less precise ACP $\widehat{Y}$ provides no additional value. We rigorously support these efficiency comparison results by carefully analyzing and comparing the asymptotic variances of the estimators using closed-form formulas.

Our contributions also include the development of semi-parametrically efficient and doubly robust estimators for the target parameter in both the unlabeled data population and the combined data population. By utilizing the cross-fitting technique, the proposed estimator achievesdouble robustness and, more importantly, attains the best possible estimation efficiency among all regular and asymptotically linear estimators.

## 2. Setup

**Semi-supervised learning setting under covariate shift.** We adopt the standard semi-supervised learning setting where we have labeled data $\mathcal{L}$ that contains independent and identically distributed (i.i.d.) $(y_i, \mathbf{x}_i)$, $i = 1, \ldots, n$, as well as unlabeled data $\mathcal{U}$ that contains i.i.d. $\mathbf{x}_j$, $j = n+1, \cdots, n+N \equiv M$. We use a binary indicator $R$ to denote whether one particular subject is from $\mathcal{L}$ ($R = 1$) or

*Table 1.* Data structure considered in this paper: Scenario I (w.o. ACP) vs Scenario II (w. ACP).

| | | SCENARIO I | | | SCENARIO II | | | |
| --- | --- | --- | --- | --- | --- | --- | --- | --- |
| | | R | Y | X | R | Y | X | $\widehat{Y}$ |
| | 1 | 1 | $\checkmark$ | $\checkmark$ | 1 | $\checkmark$ | $\checkmark$ | $\checkmark$ |
| $\mathcal{L}$ | $\vdots$ | $\vdots$ | | $\checkmark$ | $\checkmark$ | $\vdots$ | $\checkmark$ | $\checkmark$ | $\checkmark$ |
| | $n$ | 1 | $\checkmark$ | $\checkmark$ | 1 | $\checkmark$ | $\checkmark$ | $\checkmark$ |
| | $n+1$ | 0 | | $\checkmark$ | 0 | | $\checkmark$ | $\checkmark$ |
| $\mathcal{U}$ | $\vdots$ | $\vdots$ | | $\checkmark$ | $\vdots$ | | $\checkmark$ | $\checkmark$ |
| | $n+N \equiv M$ | 0 | | $\checkmark$ | 0 | | $\checkmark$ | $\checkmark$ |

from $\mathcal{U}$ ($R = 0$). Please refer to Scenario I (w.o. ACP) in Table 1 for the data structure.

We use $p(\cdot)$ and $p(\cdot \mid \cdot)$ to denote the generic marginal and conditional distributions for the labeled data $\mathcal{L}$, and $q(\cdot)$ and $q(\cdot \mid \cdot)$ for the unlabeled data $\mathcal{U}$. Instead of assuming both sources of data follow exactly the same distribution, we only assume the conditional distribution of outcome $Y$ given feature $\mathbf{X}$ remains the same, i.e., $p(y \mid \mathbf{x}) = q(y \mid \mathbf{x})$, while allowing the marginal distributions of $\mathbf{X}$ to differ between $\mathcal{L}$ and $\mathcal{U}$, i.e.,

$$p(\mathbf{x}) \neq q(\mathbf{x}).$$

This covariate shift assumption implies the independence between $Y$ and $R$ conditional on $\mathbf{X}$. That is,

$$\mathrm{pr}(R = 1 \mid Y, \mathbf{X}) = \mathrm{pr}(R = 1 \mid \mathbf{X}), \tag{1}$$

which is less stringent than the assumption that the sampling of the labeled data is purely random; i.e., $\mathrm{pr}(R = 1 \mid Y, \mathbf{X}) = \mathrm{pr}(R = 1)$.

**Objectives and metrics.** Under the covariate shift assumption, the distributions of the labeled data $\mathcal{L}$, of the unlabeled data $\mathcal{U}$ and of the combined data $\mathcal{L} \cup \mathcal{U}$ are all different. In the main paper, we focus on the $d$-dimensional parameter $\boldsymbol{\beta}$ for some characteristic in the unlabeled data population $\mathcal{U}$, defined as

$$\boldsymbol{\beta} = \mathrm{argmin}\, \mathrm{E}_q\{\ell(y, \mathbf{x}; \boldsymbol{\beta})\},$$

where $\mathrm{E}_q$ represents the expectation with respect to the distribution of $\mathcal{U}$, and $\ell(\cdot)$ denotes a generic loss function. Equivalently, with a smooth loss function, we write $\boldsymbol{\beta}$ as the solution of the estimating equation

$$\mathrm{E}_q\{\mathbf{s}(Y, \mathbf{X}; \boldsymbol{\beta})\} = \mathbf{0}. \tag{2}$$

For example, when the outcome mean $\mathrm{E}_q(Y)$ is of interest, one can define $\ell(y, \mathbf{x}; \boldsymbol{\beta})$ as $(y - \beta)^2/2$ and the corresponding $\mathbf{s}(Y, \mathbf{X}; \boldsymbol{\beta})$ is $Y - \beta$. As another example,

if the linear regression coefficient is of interest, one can define $\ell(y, \mathbf{x}; \boldsymbol{\beta})$ as $(y - \mathbf{x}^{\mathrm{T}}\boldsymbol{\beta})^2/2$ and, correspondingly, $\mathbf{s}(Y, \mathbf{X}; \boldsymbol{\beta}) = (Y - \mathbf{X}^{\mathrm{T}}\boldsymbol{\beta})\mathbf{X}$.

The overarching goal of this paper is to understand how to *best* estimate the parameter $\boldsymbol{\beta}$ without incorporating ACPs, with incorporating ACPs (below) and their comparison on estimation efficiency.

As an alternative, if some characteristic of the combined data population $\mathcal{L} \cup \mathcal{U}$ is of interest, one can define the parameter $\boldsymbol{\theta}$, similarly as above. Indeed, all the results presented in this paper for $\boldsymbol{\beta}$ can be similarly developed for $\boldsymbol{\theta}$. For the interest of space, we only present some brief results for $\boldsymbol{\theta}$ in Appendix B.

**Incorporating ACPs.** We assume that there exists a black-box prediction model, and we have automated computational phenotypes (ACPs) $\widehat{y}_i$ for every subject $i \in \{1, \cdots, M\}$. Here, this black-box model is usually trained from machine learning, natural language processing, or large language model (LLM) such as ChatGPT, where the input feature might contain not only $\mathbf{X}$ but also some other variable $\mathbf{Z}$. For the data structure, please refer to Scenario II (w. ACP) in Table 1.

To understand the statistical benefits of these ACPs in the presence of covariate shift, we further assume

$$p(\widehat{y} \mid \mathbf{x}, y) = q(\widehat{y} \mid \mathbf{x}, y).$$

Together with the original covariate shift assumption in that $p(y \mid \mathbf{x}) = q(y \mid \mathbf{x})$, it implies

$$\mathrm{pr}(R = 1 \mid Y, \widehat{Y}, \mathbf{X}) = \mathrm{pr}(R = 1 \mid \mathbf{X}). \tag{3}$$

Again, this assumption is still less stringent than the assumption that the sampling of the labeled data is purely random.

*Remark* 2.1. The assumption (3) is equivalent to stating that, $(Y, \widehat{Y})$ and $R$ are conditional independent, given $\mathbf{X}$. It can be tested when $Y$ is available in the unlabeled data but cannot if otherwise. When it cannot be tested, one can decompose the assumption as $p(\widehat{y}|\mathbf{x}) = q(\widehat{y}|\mathbf{x})$ and $p(y|\mathbf{x}, \widehat{y}) = q(y|\mathbf{x}, \widehat{y})$. It is clear that, the untestable part $p(y|\mathbf{x}, \widehat{y}) = q(y|\mathbf{x}, \widehat{y})$ is something similar to the covariate shift assumption $p(y|\mathbf{x}) = q(y|\mathbf{x})$, which is, albeit untestable, popularly adopted in semi-supervised learning and distribution shift.

**Notation.** We denote $\pi \equiv n/M$, $w(\mathbf{x}) \equiv \frac{q(\mathbf{x})}{p(\mathbf{x})}$ and $\pi(\mathbf{x}) \equiv \mathrm{pr}(R = 1 \mid \mathbf{x})$. Clearly, $w(\mathbf{x}) = \frac{\pi}{1-\pi}\frac{1-\pi(\mathbf{x})}{\pi(\mathbf{x})}$ and $\frac{1-\pi(\mathbf{x})}{1-\pi} = \frac{w(\mathbf{x})}{\pi + (1-\pi)w(\mathbf{x})}$. We denote $\mathbf{m}(\mathbf{x}) \equiv \mathrm{E}\{\mathbf{s}(Y, \mathbf{x}; \boldsymbol{\beta})|\mathbf{x}\}$ and $\widetilde{\mathbf{m}}(\mathbf{x}, \widehat{y}) \equiv \mathrm{E}\{\mathbf{s}(Y, \mathbf{x}; \boldsymbol{\beta})|\mathbf{x}, \widehat{y}\}$. We also simply write $\rho = (w(\mathbf{x}), \mathbf{m}(\mathbf{x}), \widetilde{\mathbf{m}}(\mathbf{x}, \widehat{y}))$. We use subscript $_0$ to denote the ground truth of nuisance functions and superscript

$^*$ to denote the best approximation of the truth within a possibly misspecified model, e.g., $\mathbf{m}_0(\mathbf{x})$ and $\mathbf{m}^*(\mathbf{x})$. We assume the $d \times d$ symmetric matrix $\mathrm{E}_q\{\partial\mathbf{s}(Y, \mathbf{X}; \boldsymbol{\beta})/\partial\boldsymbol{\beta}^{\mathrm{T}}\}$ evaluated at the true $\boldsymbol{\beta}_0$ is invertible and denote this inverse as $\boldsymbol{\Omega}$. Clearly, $\boldsymbol{\Omega} = \boldsymbol{\Omega}^{\mathrm{T}}$. For random vector $\boldsymbol{\phi}$, we write $\mathrm{E}(\boldsymbol{\phi}\boldsymbol{\phi}^{\mathrm{T}})$ as $\mathrm{E}(\boldsymbol{\phi}^{\otimes 2})$.

## 3. Efficiency Gain Analysis

In this section, we mainly analyze the statistical benefits, in the sense of estimation efficiency, of the ACP $\widehat{y}_i$'s. Our first step is to understand the optimal estimation efficiency with and without incorporating these $\widehat{y}_i$'s, for estimating $\boldsymbol{\beta}$. Here, the optimal estimation efficiency is the so-called semi-parametric efficiency bound, characterized by the efficient influence function, also referred to as canonical gradient, or, efficient influence curve (Bickel et al., 1993; Tsiatis, 2006; Fisher & Kennedy, 2021; Hines et al., 2022; Ichimura & Newey, 2022).

### 3.1. Optimal Efficiency Bound with and without ACPs

We first briefly introduce the asymptotic representation of a regular asymptotically linear (RAL) estimator. In general, given i.i.d. copies of the random sample $\{\mathbf{d}_1, \ldots, \mathbf{d}_n\}$ with sample size $n$, an estimator for the parameter of interest $\boldsymbol{\beta}$, $\widehat{\boldsymbol{\beta}}$, is a RAL estimator if

$$\sqrt{n}(\widehat{\boldsymbol{\beta}} - \boldsymbol{\beta}) = n^{-1/2}\sum_{i=1}^{n}\boldsymbol{\phi}(\mathbf{d}_i) + o_p(1),$$

where the zero-mean function $\boldsymbol{\phi}(\cdot)$ is called the influence function of $\widehat{\boldsymbol{\beta}}$, and the corresponding asymptotic variance is $\mathrm{E}(\boldsymbol{\phi}\boldsymbol{\phi}^{\mathrm{T}})$, provided that it is finite and nonsingular. Among all RAL estimators for $\boldsymbol{\beta}$, the influence function of the one with the smallest asymptotic variance is called the efficient influence function (EIF), $\boldsymbol{\phi}_{\mathrm{EIF}}$, and the optimal efficiency bound is $\mathrm{E}(\boldsymbol{\phi}_{\mathrm{EIF}}\boldsymbol{\phi}_{\mathrm{EIF}}^{\mathrm{T}})$.

To proceed, we first consider the situation with the ACPs, Scenario II in Table 1. The result below, Proposition 3.1, presents the EIF for estimating $\boldsymbol{\beta}$ under this situation, as well as the efficiency bound. The proof of this result is contained in Appendix A. The main idea is to first derive the semiparametric tangent space $\mathcal{T}$ (Bickel et al., 1993; Tsiatis, 2006) under this situation, that is defined as the mean squared closure of the tangent spaces of all parametric submodels spanned by the score vectors, and then derive the EIF using the orthogonality.

**Proposition 3.1.** *With the ACP $\widehat{y}_i$'s, the EIF for estimating $\boldsymbol{\beta}$ defined in* (2) *is $\boldsymbol{\Omega}\boldsymbol{\phi}_w$ with $\boldsymbol{\phi}_w$ equals*

$$\frac{r}{\pi}w_0(\mathbf{x})\{\mathbf{s}(y, \mathbf{x}; \boldsymbol{\beta}_0) - \widetilde{\mathbf{m}}_0(\mathbf{x}, \widehat{y})\} + \frac{1-r}{1-\pi}\mathbf{m}_0(\mathbf{x})$$

$$+ \frac{w_0(\mathbf{x})}{\pi + (1-\pi)w_0(\mathbf{x})}\{\widetilde{\mathbf{m}}_0(\mathbf{x}, \widehat{y}) - \mathbf{m}_0(\mathbf{x})\}, \tag{4}$$

*and the efficiency bound equals* $\boldsymbol{\Omega}\mathbf{V}_w\boldsymbol{\Omega}$, *where* $\mathbf{V}_w$ *is*

$$\frac{1}{\pi}E_p\left[w_0(\mathbf{X})^2\{\mathbf{s}(Y,\mathbf{X};\boldsymbol{\beta}_0)-\widetilde{\mathbf{m}}_0(\mathbf{X},\widehat{Y})\}^{\otimes 2}\right]$$
$$+\frac{1}{(1-\pi)^2}E\left[\{1-\pi_0(\mathbf{X})\}^2\{\widetilde{\mathbf{m}}_0(\mathbf{X},\widehat{Y})-\mathbf{m}_0(\mathbf{X})\}^{\otimes 2}\right]$$
$$+\frac{1}{1-\pi}E_q\{\mathbf{m}_0(\mathbf{X})^{\otimes 2}\}.$$

Next we derive the efficiency bound for Scenario I in Table 1, without the ACPs. This can be regarded as a special case of Scenario II in Table 1, with the ACPs, so the proof is omitted.

**Proposition 3.2.** *Without* $\widehat{y}_i$'s, *the EIF for estimating* $\boldsymbol{\beta}$ *defined in* (2) *is* $\boldsymbol{\Omega}\phi_{wo}$ *where*

$$\phi_{wo}=\frac{r}{\pi}w_0(\mathbf{x})\{\mathbf{s}(y,\mathbf{x};\boldsymbol{\beta}_0)-\mathbf{m}_0(\mathbf{x})\}+\frac{1-r}{1-\pi}\mathbf{m}_0(\mathbf{x}),$$

*and the efficiency bound equals* $\boldsymbol{\Omega}\mathbf{V}_{wo}\boldsymbol{\Omega}$, *where*

$$\mathbf{V}_{wo}=\frac{1}{\pi}E_p\left[w_0^2(\mathbf{X})\{\mathbf{s}(Y,\mathbf{X};\boldsymbol{\beta}_0)-\mathbf{m}_0(\mathbf{X})\}^{\otimes 2}\right]$$
$$+\frac{1}{1-\pi}E_q\left\{\mathbf{m}_0(\mathbf{X})^{\otimes 2}\right\}.$$

### 3.2. Efficiency Gain and its Source

To further understand the efficiency gain of the ACPs, which is the contrast between Scenario I and Scenario II in Table 1, we have

**Proposition 3.3.** *The difference of the two asymptotic variances,* $\boldsymbol{\Omega}\mathbf{V}_{wo}\boldsymbol{\Omega}-\boldsymbol{\Omega}\mathbf{V}_w\boldsymbol{\Omega}$, *equals*

$$\frac{1}{(1-\pi)^2}\boldsymbol{\Omega}E\left[\frac{\{1-\pi_0(\mathbf{X})\}^3}{\pi_0(\mathbf{X})}\{\widetilde{\mathbf{m}}_0(\mathbf{X},\widehat{Y})-\mathbf{m}_0(\mathbf{X})\}^{\otimes 2}\right]\boldsymbol{\Omega},$$

*which is always positive semidefinite.*

*Remark* 3.4. The proof of Proposition 3.3 is contained in Appendix A. From Proposition 3.3, it is clear that incorporating ACPs will not attenuate the estimation efficiency. The extreme special case is that, the generation of $\widehat{Y}$ only depends on the available feature $\mathbf{X}$; that is, $\widehat{Y}$ is a function of $\mathbf{X}$. In this case, one can derive that $\widetilde{\mathbf{m}}_0(\mathbf{x},\widehat{y})=\mathbf{m}_0(\mathbf{x})$ and there is no efficiency gain. The intuition is that, in this case, $\widehat{Y}$ does not bring in any *new* information beyond $\mathbf{X}$ and thus does not bring in efficiency gain. If the generation of $\widehat{Y}$ depends on $\mathbf{X}$ as well as some other different variable $\mathbf{Z}$ predictive of $Y$, the estimator with ACPs does bring positive efficiency gain, compared to the estimator without ACPs.

So, where exactly does the efficiency gain come from? The ACPs for the labeled data, or the ACPs for the unlabeled data, or both? To answer this question, we consider an intermediate scenario as shown in Table 2 below. For this scenario, we have the following

*Table 2.* Data structure considered in this paper: Scenario III (intermediate scenario between Scenario I and Scenario II).

| | | R | Y | X | $\widehat{Y}$ |
|---|---|---|---|---|---|
| | | SCENARIO III | | | |
| $\mathcal{L}$ | 1 | 1 | $\checkmark$ | $\checkmark$ | $\checkmark$ |
| | $\vdots$ | $\vdots$ | $\checkmark$ | $\checkmark$ | $\checkmark$ |
| | $n$ | 1 | $\checkmark$ | $\checkmark$ | $\checkmark$ |
| $\mathcal{U}$ | $n+1$ | 0 | | | $\checkmark$ |
| | $\vdots$ | $\vdots$ | | | $\checkmark$ |
| | $n+N\equiv M$ | 0 | | | $\checkmark$ |

**Proposition 3.5.** *For the intermediate scenario in Table 2, the estimation efficiency bound equals to the one in Proposition 3.2, corresponding to Scenario I in Table 1.*

*Remark* 3.6. The proof of Proposition 3.5 is similar to Proposition 3.1 so omitted. It implies, if the ACPs for the labeled data only were available, then this set of ACPs does not bring in any efficiency gain. Therefore, compared to Scenario I in Table 1, the efficiency gain of Scenario II in Table 1 comes from the ACPs for the unlabeled data. This is intuitive. For the labeled data, the ground truth $Y$'s exist and thus the ACPs $\widehat{Y}$'s do not contribute; while for the unlabeled data, the ground truth $Y$'s do not exist and their ACPs $\widehat{Y}$'s do contribute.

## 4. Estimation

We mainly present the estimator $\widehat{\boldsymbol{\beta}}$ that incorporates the ACPs. From the EIF presented in (4), to construct an efficient estimator, one needs to estimate the relevant nuisance functions $\rho=(w(\mathbf{x}),\widetilde{\mathbf{m}}(\mathbf{x},\widehat{y}),\mathbf{m}(\mathbf{x}))$. To accommodate estimations of nuisance functions using nonparametric or

---

**Algorithm 1** Estimator and Confidence Interval

**Input:** Labeled dataset $D^l=\{(\mathbf{x}_i,y_i,\widehat{y}_i):i=1,\ldots,n\}$, unlabeled dataset $D^u=\{(\mathbf{x}_i,\widehat{y}_i):i=n+1,\ldots,n+N\}$. To use cross-fitting, define $D^l=D_1^l\cup\cdots\cup D_K^l$ and $D^u=D_1^u\cup\cdots\cup D_K^u$, with $|D_1^l|=\cdots=|D_K^l|=n/K$ and $|D_1^u|=\cdots=|D_K^u|=N/K$. Define $D_k=D_k^l\cup D_k^u,k=1,\cdots,K$.
**Output:** Obtain $\widehat{\boldsymbol{\beta}}$ and confidence interval of $\mathbf{v}^{\mathrm{T}}\boldsymbol{\beta}$ for any vector $\mathbf{v}\in\mathbb{R}^d$.
**for** $k=1$ **to** $K$ **do**
    Estimate $\rho_k$ using $D_k^c$, obtain $\widehat{\rho}_k$.
**end for**
Plug $\widehat{\rho}$ into equation (5) and solve the equation, obtain $\widehat{\boldsymbol{\beta}}$.
Plug $\widehat{\rho}$ and $\widehat{\boldsymbol{\beta}}$ into equation (6) and obtain $\widehat{\boldsymbol{\Omega}}\widehat{\mathbf{V}}_w\widehat{\boldsymbol{\Omega}}$.
Obtain the confidence intervals of $\mathbf{v}^{\mathrm{T}}\boldsymbol{\beta}$ by equation (8).

---

machine learning methods, we adopt the cross-fitting technique (Chernozhukov et al., 2018) in our estimation.

Take K-fold random partitions $\{D_k^l\}_{k=1}^K$ and $\{D_k^u\}_{k=1}^K$ of the labeled and unlabeled data sets $D^l$ and $D^u$, respectively. Then $D_k = D_k^l \cup D_k^u$ constitutes a K-fold random partition of the whole data set $\{(\mathbf{x}_i, r_i y_i, \widehat{y}_i)_{i=1}^M\}$. For each $k = 1, \ldots, K$, we define $D_k^c = \{(\mathbf{x}_i, r_i y_i, \widehat{y}_i)_{i=1}^M\}/D_k$ and let $\widehat{\rho}_k = (\widehat{w}_k(\mathbf{x}), \widehat{\widetilde{\mathbf{m}}}_k(\mathbf{x}, \widehat{y}), \widehat{\mathbf{m}}_k(\mathbf{x}))$ denote the estimates of nuisance functions obtained using the data set $D_k^c$ only. Let $\widehat{\rho} = (\widehat{\rho}_1, \ldots, \widehat{\rho}_K)$. Then the estimator $\widehat{\boldsymbol{\beta}}(\widehat{\rho})$, shorthanded as $\widehat{\boldsymbol{\beta}}$, is the solution of the following equation:

$$
\frac{1}{K} \sum_{k=1}^K \widehat{E}_k \left[ \frac{R}{\pi} \widehat{w}_k(\mathbf{X}) \{ \mathbf{s}(Y, \mathbf{X}; \widehat{\boldsymbol{\beta}}) - \widehat{\widetilde{\mathbf{m}}}_k(\mathbf{X}, \widehat{Y}) \} \right.
$$
$$
+ \frac{\widehat{w}_k(\mathbf{X})}{\pi + (1-\pi)\widehat{w}_k(\mathbf{X})} \{ \widehat{\widetilde{\mathbf{m}}}_k(\mathbf{X}, \widehat{Y}) - \widehat{\mathbf{m}}_k(\mathbf{X}) \}
$$
$$
\left. + \frac{1-R}{1-\pi} \widehat{\mathbf{m}}_k(\mathbf{X}) \right] = \mathbf{0}, \tag{5}
$$

where $\widehat{E}_k$ denotes the sample average over the k-th fold, and we denote the left-hand side of the above equation as $\mathbf{N}(\widehat{\rho}, \widehat{\boldsymbol{\beta}})$. We briefly summarize the computational algorithm in Algorithm 1.

*Remark* 4.1. In our implementation, for the estimations of nuisance functions, we adopted super learner (Van der Laan et al., 2007), an ensemble learning approach that improves estimation by combining multiple machine learning models with data-adaptive weights.

## 5. Theoretical Investigations

Now we develop the theoretical results for the proposed estimator $\widehat{\boldsymbol{\beta}}$, with all the proofs contained in Appendix A. Firstly, we need some assumptions on the nuisance functions and their estimates.

**Assumption 5.1** (Requirements on Nuisance Estimators). For $k = 1, \ldots, K$, we assume positive density ratio models such that $\widehat{w}_k(\mathbf{x})$ and $w^*(\mathbf{x})$ are bounded away from zero. We assume the nuisance estimators $\widehat{\rho}_k = (\widehat{w}_k(\mathbf{x}), \widehat{\widetilde{\mathbf{m}}}_k(\mathbf{x}, \widehat{y}), \widehat{\mathbf{m}}_k(\mathbf{x}))$ converges to the limit $\rho^* = (w^*(\mathbf{x}), \widetilde{\mathbf{m}}^*(\mathbf{x}, \widehat{y}), \mathbf{m}^*(\mathbf{x}))$ in MSE at the following rates:

$$
\|\widehat{w}_k(\mathbf{x}) - w^*(\mathbf{x})\|_2 = a_{1M},
$$
$$
\|\widehat{\mathbf{m}}_k(\mathbf{x}) - \mathbf{m}^*(\mathbf{x})\|_2 = a_{2M},
$$
$$
\|\widehat{\widetilde{\mathbf{m}}}_k(\mathbf{x}, \widehat{y}) - \widetilde{\mathbf{m}}^*(\mathbf{x}, \widehat{y})\|_2 = a_{3M},
$$

where $\max\{a_{1M}, a_{2M}, a_{3M}\} \to 0$ as $M \to +\infty$.

*Remark* 5.2. The regularity conditions on nuisance estimators are standard and widely advocated in double/debiased ML literature, e.g., Van der Laan & Rose (2011), Chernozhukov et al. (2018), Kennedy (2024), Chernozhukov

et al. (2024). The convergence rate is achievable for many ML methods such as in regression trees and random forests (Wager & Walther, 2015) and a class of neural nets (Chen & White, 1999). One can refer to Chernozhukov et al. (2018) for more examples.

**Theorem 5.3** (Double Robustness). *Assume Assumption 5.1 holds. If either $w^*(\mathbf{x}) = w_0(\mathbf{x})$ or $(\widetilde{\mathbf{m}}^*(\mathbf{x}, \widehat{y}), \mathbf{m}^*(\mathbf{x})) = (\widetilde{\mathbf{m}}_0(\mathbf{x}, \widehat{y}), \mathbf{m}_0(\mathbf{x}))$, then*

$$
\|\widehat{\boldsymbol{\beta}} - \boldsymbol{\beta}_0\|_2 = o_p(1).
$$

*Remark* 5.4. Theorem 5.3 establishes the classic double robustness property, meaning that the proposed estimator $\widehat{\boldsymbol{\beta}}$ remains consistent if $w^*(\mathbf{x}) = w_0(\mathbf{x})$ or $(\widetilde{\mathbf{m}}^*(\mathbf{x}, \widehat{y}), \mathbf{m}^*(\mathbf{x})) = (\widetilde{\mathbf{m}}_0(\mathbf{x}, \widehat{y}), \mathbf{m}_0(\mathbf{x}))$.

Consequently, we can develop the asymptotic representation of the proposed estimator $\widehat{\boldsymbol{\beta}}$ and further show that it achieves the optimal efficiency bound derived in Proposition 3.1.

**Theorem 5.5** (Asymptotic Normality and Semiparametric Efficiency). *Assume Assumption 5.1 holds. Further, if we assume $a_{1M}a_{2M} = o(M^{-1/2})$, $a_{1M}a_{3M} = o(M^{-1/2})$, and that all nuisance components are correctly specified, then as $M \to \infty$,*

$$
\sqrt{M}(\widehat{\boldsymbol{\beta}} - \boldsymbol{\beta}_0) \xrightarrow{d} N(\mathbf{0}, \boldsymbol{\Omega}\mathbf{V}_w\boldsymbol{\Omega}),
$$

*where $\boldsymbol{\Omega}\mathbf{V}_w\boldsymbol{\Omega} = \boldsymbol{\Omega}E(\boldsymbol{\phi}_w^{\otimes 2})\boldsymbol{\Omega}$ is the semiparametric efficiency bound derived in Proposition 3.1.*

*Remark* 5.6. Theorem 5.5 further establishes that if all nuisance estimators converge to their true values at a sufficiently fast rate, the proposed estimator $\widehat{\boldsymbol{\beta}}$ attains a convergence rate of $O_p(M^{-1/2})$ and is asymptotically normal with the efficiency lower bound $\boldsymbol{\Omega}\mathbf{V}_w\boldsymbol{\Omega}$ as its limiting variance. Further, it introduces the concept of rate double robustness. As one can see from Theorem 5.5, what matters is the products $a_{1M}a_{2M}$ and $a_{1M}a_{3M}$, instead of the individual convergence rates $a_{1M}$, $a_{2M}$ and $a_{3M}$. Notably, this rate requirement is relatively mild, ensuring consistency and efficiency even when all nuisance estimators converge at $o_p(M^{-1/4})$, which is considerably slower than the parametric rate $O_p(M^{-1/2})$. In particular, we impose no restrictions requiring nuisance estimators to belong to Donsker or bounded entropy classes (van der Vaart, 1998), thereby permitting the use of flexible machine learning methods. Furthermore, the product rate condition allows faster-converging estimators to offset the effects of slower-converging ones, enhancing practical applicability.

*Remark* 5.7. An interesting scenario is when we have a lot more unlabeled data in that $N \gg n$ so $\pi \to 0$. Accordingly, one can derive that $\boldsymbol{\Omega}\mathbf{V}_w\boldsymbol{\Omega} = O(\pi^{-1})$ instead of $O(1)$. Thus, the above asymptotic representation shall be more precisely written as

$$
\sqrt{n}(\widehat{\boldsymbol{\beta}} - \boldsymbol{\beta}_0) \xrightarrow{d} N(\mathbf{0}, \pi\boldsymbol{\Omega}\mathbf{V}_w\boldsymbol{\Omega}).
$$

This indicates, even if we have sufficiently many unlabeled data, the *real* sample size for deriving the efficient estimator is still $n$. Intuitively we do not have the ground truth $Y$ in the unlabeled data, so they (with sample size $N$) cannot contribute to increasing the convergence rate.

Finally we present how to construct the confidence interval for the linear combination $\mathbf{v}^{\mathrm{T}}\boldsymbol{\beta}$ as well as its validity.

**Theorem 5.8** (Construction of Confidence Interval)**.** *Assume Assumption 5.1 holds. If we assume $a_{1M}a_{2M} = o(M^{-1/2})$, $a_{1M}a_{3M} = o(M^{-1/2})$, and that all nuisance components are correctly specified, then as $M \to \infty$,*

$$\widehat{\boldsymbol{\Omega}}\widehat{\mathbf{V}}_w\widehat{\boldsymbol{\Omega}} = \widehat{\boldsymbol{\Omega}}\frac{1}{K}\sum_{k=1}^{K}\widehat{E}_k\{\phi_w^{\otimes 2}(\widehat{\rho};\widehat{\boldsymbol{\beta}})\}\widehat{\boldsymbol{\Omega}} \xrightarrow{p} \boldsymbol{\Omega}\mathbf{V}_w\boldsymbol{\Omega}. \quad (6)$$

*Consequently, for any vector $\mathbf{v} \in \mathbb{R}^d$, we have*

$$\frac{\sqrt{M}\mathbf{v}^{\mathrm{T}}(\widehat{\boldsymbol{\beta}} - \boldsymbol{\beta}_0)}{(\mathbf{v}^{\mathrm{T}}\widehat{\boldsymbol{\Omega}}\widehat{\mathbf{V}}_w\widehat{\boldsymbol{\Omega}}\mathbf{v})^{1/2}} \xrightarrow{d} N(0,1). \quad (7)$$

*Thus, we can construct a $(1-\alpha)\times 100\%$ confidence interval for $\gamma = \mathbf{v}^{\mathrm{T}}\boldsymbol{\beta}_0$,*

$$\mathrm{CI}_\alpha = \{\gamma : |\gamma - \mathbf{v}^{\mathrm{T}}\widehat{\boldsymbol{\beta}}| \leq \Phi^{-1}(1-\alpha/2)M^{-1/2}\widehat{\mathrm{sd}}\}, \quad (8)$$

*where $\widehat{\mathrm{sd}} = (\mathbf{v}^{\mathrm{T}}\widehat{\boldsymbol{\Omega}}\widehat{\mathbf{V}}_w\widehat{\boldsymbol{\Omega}}\mathbf{v})^{1/2}$ and $\Phi$ is the cumulative distribution function of the standard normal distribution. Further, this CI satisfies*

$$\lim_{M \to \infty} pr\left(\mathbf{v}^{\mathrm{T}}\boldsymbol{\beta}_0 \in \mathrm{CI}_\alpha\right) = 1 - \alpha. \quad (9)$$

*Remark* 5.9. Theorem 5.8 establishes that, under the same conditions, the efficiency lower bound can be consistently estimated by constructing the sample estimator of $\mathrm{E}\{\phi_w^{\otimes 2}(\rho_0, \boldsymbol{\beta}_0)\}$ using cross-fitted nuisance estimators and the proposed estimator $\widehat{\boldsymbol{\beta}}$. Consequently, the confidence interval in equation (8) asymptotically attains the correct coverage probability. Moreover, since the proposed estimator $\mathbf{v}^{\mathrm{T}}\widehat{\boldsymbol{\beta}}$ asymptotically achieves the smallest possible variance, the confidence interval in equation (8) tends to be shorter than those based on less efficient estimators.

## 6. Numerical Evaluations

### 6.1. Synthetic Data

In this section, we demonstrate the efficiency gains achieved by incorporating ACPs into the estimation process compared to the estimation without ACPs through simulation studies.

The simulation setup is as follows. Recall that the total sample size from the labeled and unlabeled data is $M = n + N$, and here we vary $n, N \in \{300, 600, 900, 1200, 1500\}$. The covariate vector $\mathbf{X}$ has a dimension of $p = 5$, and we generate an additional variable $Z$ which is used as the ACP

*Figure 1.* Variation of ARE for the estimation of $\mathrm{E}_q(Y)$ under different sample sizes, signal strength, and correlation coefficient.

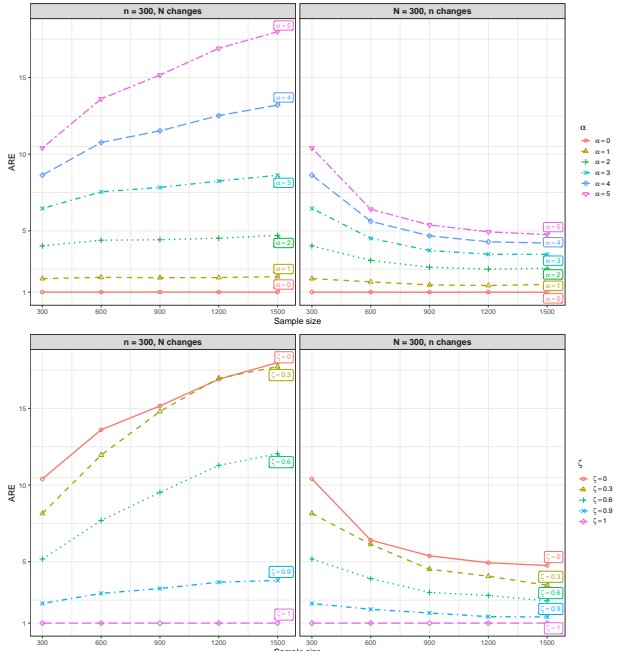

$\widehat{Y}$. The vector $(\mathbf{X}, Z)$ follows a multivariate normal distribution $N(\mathbf{0}_{p+1}, \boldsymbol{\Sigma})$. The covariance matrix $\boldsymbol{\Sigma}$ is given by $\boldsymbol{\Sigma} = \mathbf{I}_{p+1} + \boldsymbol{\Lambda}$, where $\boldsymbol{\Lambda}_{1(p+1)} = \boldsymbol{\Lambda}_{(p+1)1} = \zeta$, and all other elements are zero. We vary $\zeta$ in $\{0, 0.3, 0.6, 0.9, 1\}$ to characterize how much information in $\widehat{Y}$ is overlapped with $\mathbf{X}$. Two regression models are considered: for continuous outcomes, we set $Y_i = 1 + \boldsymbol{\xi}^{\mathrm{T}}\mathbf{X}_i + \alpha Z_i + \varepsilon_i$, where $\varepsilon_i \sim N(0,1)$, while for binary outcomes, we set $Y_i \sim \mathrm{Bernoulli}\{\mathrm{logit}^{-1}(1 + \boldsymbol{\xi}^{\mathrm{T}}\mathbf{X}_i + \alpha Z_i)\}$ for all $i = 1, \ldots, n + N$. In both cases, $\boldsymbol{\xi} = (1, \mathbf{0.5}_4)^{\mathrm{T}}$ and $\alpha$ is varied in $\{0, 1, \ldots, 5\}$ to represent varying accuracy of the ACP. Additionally, we generate a binary variable $R_i \sim \mathrm{Bernoulli}\{\mathrm{logit}^{-1}(\boldsymbol{\eta}^{\mathrm{T}}\mathbf{X}_i)\}$ and $\boldsymbol{\eta} = \mathbf{1}_5^{\mathrm{T}}$. $R_i = 1$ indicates labeled data with $Y_i$ observed, whereas $R_i = 0$ indicates unlabeled data with $Y_i$ unavailable.

For the parameters of interest, we consider the mean of $Y$ on the unlabeled dataset $E_q(Y)$ and the regression coefficients solving the estimating equation $E_q[\{Y - g(\mathbf{X}^{\mathrm{T}}\boldsymbol{\beta})\}\mathbf{X}] = \mathbf{0}$ where $g(\cdot)$ is the identify function for continuous outcomes and the expit function for binary outcomes. To compare the performance of the estimators with and without $\widehat{Y}$, we calculate their MSEs based on 500 simulation replications.

We report the results on $\mathrm{E}_q(Y)$ in linear models in Figure 1, while deferring all other results to Appendix C. Specifically, Figure 1 shows the efficiency improvement by incorporating ACPs, under varying values of labeled sample size ($n$), unlabeled sample size ($N$), signal strength from the ACP

*Figure 2.* Difference in distributions of the inpatient visit count for labeled (left) and unlabeled (right) dataset.

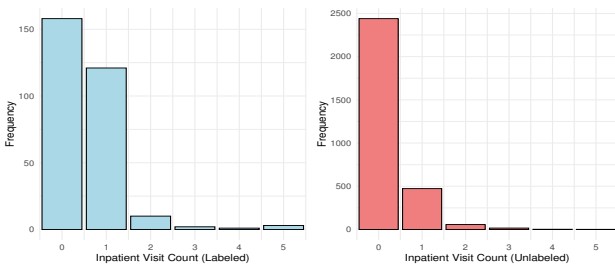

($\alpha$), and the correlation between ACP and the covariates ($\zeta$). Efficiency gain is quantified by Asymptotic Relative Efficiency (ARE), defined as the ratio of the MSE of the estimate without ACP to that with ACP.

First, we note that ARE greater than 1 indicates incorporating ACPs results in smaller MSE and thus, more efficient estimation. This is indeed the case across all simulation settings, except when the ACP is not predictive of the true outcome ($\alpha = 0$) or when it provides no additional information beyond what is captured in $\mathbf{X}$ ($\zeta = 1$).

Second, ARE generally increases as $\zeta$ decreases, with the maximum ARE achieved at $\zeta = 0$. Note that $\zeta = 0$ indicates that $\widehat{Y}$ is independent from the predictors, and thus the ACP provides the most additional information. We also observe that ARE increases with increasing $\alpha$, as the ACP becomes more predictive of the true outcome. Given fixed values of $\alpha$ and $\zeta$, when the sample size of the unlabeled dataset increases, the ARE generally increases, although the rate is relatively moderate; when the sample size of the labeled dataset increases, the ARE exhibits a quadratic decreasing trend before gradually stabilizing, aligning with the results in our theoretical investigations.

### 6.2. Diabetes Data

In this section, we implement our proposed method in our motivating study, to identify risk factors for diabetes among children and adolescents.

The study population comprised individuals under 18 years of age who had at least one encounter recorded between January 1, 2012, and December 31, 2020. Using the criteria outlined in Li et al. (2025), we identified 3,000 patients who are suspected to have diabetes, including both Type I and Type II diabetes, as our unlabeled dataset. For these patients, we followed an ACP using a decision-tree-based algorithm to compute and identify diabetes status. Besides, through stratified random sampling, 297 patients were selected, and their documented visit summary and EHRs were sent to and reviewed by medical experts who provided binary (yes/no) assessments of diabetes status. We refer to this dataset as

*Figure 3.* The 95% confidence interval for the coefficient corresponding to each variable. Heavy comorbidity burden is defined as CCI$\geq 2$.

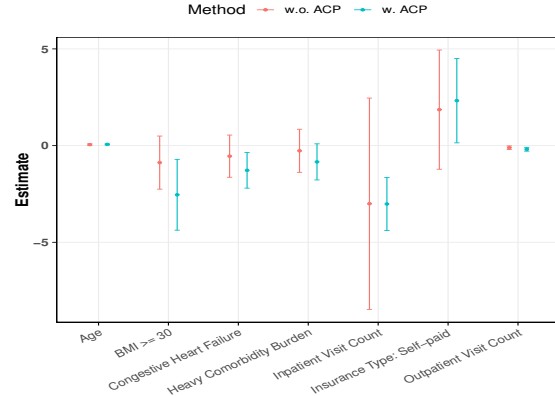

our labeled dataset. Due to the stratified random sampling, we observed significant covariate shifts between the labeled and unlabeled datasets. Figure 2 provides strong evidence that the distribution of inpatient visit counts differs substantially. Patients in chart review are more likely to have more inpatient visit counts, which may be due to the need to have sufficient patient information for chart review. Other available features include socio-demographic variables, e.g., sex and race/ethnicity, and clinical variables, e.g., family history of diabetes and Charlson Comorbidity Index (CCI) scores. A cohort summary associated with socio-demographic variables is shown in Table 3, which shows that the distributions of many covariates differ substantially.

*Table 3.* Comparison of the distributions of each $X$ variable between labeled and unlabeled data: t-test is used to assess whether the means of the two groups differ significantly for continuous variables, and chi-square test is used to evaluate whether the distributions between the two groups are the same for categorical variables. **A p-value less than 0.05** indicates the statistical significance.

| Variable | Labeled ($n = 297$) | Unlabeled ($N = 3,000$) | p-value |
|---|---|---|---|
| Age | 7.59 ± 6.46 | 7.42 ± 5.35 | 0.653 |
| Inpatient Visit Count | 0.6 ± 0.89 | 0.25 ± 0.71 | < 0.001 |
| Outpatient Visit Count | 3.43 ± 6.67 | 2.7 ± 5.18 | 0.070 |
| Insurance Type: Self-Paid | 24 (8.1%) | 59 (2%) | < 0.001 |
| BMI$\geq$ 30 (Yes) | 59 (19.9%) | 192 (6.4%) | < 0.001 |
| Congestive Heart Failure (Yes) | 28 (9.4%) | 85 (2.8%) | < 0.001 |
| Heavy Comorbidity Burden | 20 (6.7%) | 85 (2.8%) | < 0.001 |

To identify driving factors for diabetes diagnosis, we first conducted marginal screening of all categorical variables using Fisher's exact test. Variables with a p-value $< 0.05$, adjusted by the Benjamini-Hochberg procedure to control the false discovery rate, were selected. We adopted a logistic regression to predict diabetes status with all selected variables, and use the regression coefficients as association measures. We then implemented the proposed methods with and with-

out ACPs to estimate these coefficients. For nuisance parameter estimation, we employed super learner implemented via the R package `SuperLearner` with the following libraries: generalized linear models, random forests, kernel support vector machines, and XGBoost. These algorithms were individually tuned using 10-fold cross-validation. To accommodate the flexible estimations for nuisance parameters, we also applied a cross-fitting algorithm. To obtain confidence intervals associated with the target parameters, a perturbed bootstrap procedure with 500 repetitions was implemented. In each repetition, individual sample weights were generated from an independent $\text{Exp}(1)$ distribution. Finally, the confidence interval was constructed using the 0.025 and 0.975 quantiles of the 500 bootstrap repetitions.

Figure 3 summarizes the estimates of the coefficients and their 95% confidence intervals. Overall, compared with the method without ACPs, the proposed method with ACPs yields shorter intervals, which reflects the efficiency gain by incorporating ACPs in our proposed method. By incorporating ACPs, we identified insurance type (self-paid) as an important factor associated with diabetes. The self-paid patients often have limited access to preventative care and experience higher stress levels, which are known factors associated with Type 2 diabetes among adults (Kelly & Ismail, 2015; Stark Casagrande & Cowie, 2012).

### 6.3. Other Real-World Datasets

In this section we implement the proposed approach on three other data sets, and also compare with some existing methods: PPI (Angelopoulos et al., 2023a), PPI++ (Angelopoulos et al., 2023b), and RePPI (Ji et al., 2025).

**Income data:** Following Angelopoulos et al. (2023a) and Angelopoulos et al. (2023b), we analyze the relationship between wage (measured as log-income) and age, with sex as a confounding variable, using U.S. Census data under a covariate shift setting. The ACP prediction $\widehat{Y}$ is obtained by fitting an XGBoost model to log-income using 14 covariates, including education, marital status, citizenship, race, and others (Ji et al., 2025). To induce covariate shift, we partition the labeled and unlabeled datasets based on the selection probability $\exp(\alpha^{\mathrm{T}}\mathbf{X})/\{1 + \exp(\alpha^{\mathrm{T}}\mathbf{X})\}$, where $\alpha = (0,1,0)^{\mathrm{T}}$ and $\mathbf{X} = (1, X_1, X_2)^{\mathrm{T}}$, with $X_1$ denoting age and $X_2$ denoting sex. This yields a final labeled-to-unlabeled data ratio of approximately $2:8$.

**Politeness data:** Using the data from Danescu-Niculescu-Mizil et al. (2013) that comprises texts from 5,512 online requests posted on Stack Exchange and Wikipedia, we try to understand the association between politeness score (range from 1 to 25) and a binary indicator for hedging within the request (Gligorić et al., 2024). The ACP $\widehat{Y}$ is generated using OpenAI's GPT-4o mini-model that has the same range as the politeness score. To demonstrate the co-

variate shift, we split the labeled and unlabeled data in a $1:9$ ratio, following the same procedure as in the income data, where $\mathbf{X} = (1, X_1)^{\mathrm{T}}$ with $X_1$ representing hedge and $\alpha = (0,1)^{\mathrm{T}}$.

**Wine data:** Using the Wine Enthusiast review dataset, we investigate the association between wine rating (range from 80 to 100) and wine price, adjusted by wine region (Ji et al., 2025). Similar to the politeness data, the ACP $\widehat{Y}$ is also generated by employing OpenAI's GPT-4o mini-model that produces predicted ratings with the same scale. To assess covariate shift, we follow the same procedure as in the previous experiments, splitting the labeled and unlabeled data in a $3:7$ ratio. Here, $\mathbf{X} = (1, X_1, X_2, X_3, X_4, X_5)^{\mathrm{T}}$, where $X_1$ represents price, $X_2$ to $X_5$ represents California, Washington, Oregon and New York, respectively, with $\alpha = (0,1,0,0,0,0)^{\mathrm{T}}$.

For each of these three data sets, we repeat the data splitting 50 times, and then compute the length of 95% confidence intervals of the regression coefficients. We compare the proposed method with PPI, PPI++ and RePPI, with results contained in Table 4. While there are some cases that the proposed method is tiny slightly less efficient than PPI++ or RePPI, it is generally a lot more efficient than PPI, PPI++, and RePPI, with the largest reduction in confidence interval length reaching approximately 60%.

*Table 4.* Comparison of the length of 95% confidence intervals between proposed method and three alternatives: PPI, PPI++, RePPI, on three datasets: income, politeness, wine. **Bolded values (less than 1)** indicate that the proposed method has efficiency gain.

| Dataset | Variable | PPI | PPI++ | RePPI | proposed | proposed/PPI | proposed/PPI++ | proposed/RePPI |
|---|---|---|---|---|---|---|---|---|
| Income | Age | 0.0014 | 0.0012 | 0.0012 | 0.0008 | **57.14%** | **66.67%** | **66.67%** |
| | Sex | 0.0622 | 0.0541 | 0.0545 | 0.0538 | **86.51%** | **99.54%** | **98.73%** |
| Politeness | Hedge | 1.9696 | 1.7858 | 1.7399 | 1.2818 | **65.17%** | **71.70%** | **73.77%** |
| Wine | Price | 0.4007 | 0.2912 | 0.2791 | 0.1664 | **41.55%** | **57.11%** | **59.60%** |
| | California | 1.6946 | 0.7935 | 0.7693 | 0.7845 | **46.39%** | **98.87%** | 101.95% |
| | Washington | 1.7439 | 0.8436 | 0.8105 | 0.8105 | **46.49%** | **96.01%** | 100.00% |
| | Oregon | 1.8305 | 0.9057 | 0.8616 | 0.8663 | **47.35%** | **95.74%** | 100.55% |
| | New York | 1.8989 | 0.8964 | 0.8955 | 0.9174 | **48.32%** | 102.45% | 102.55% |

## 7. Discussion

In this paper, we explore the benefits of incorporating ACPs in a semi-supervised learning framework under covariate shift, particularly in terms of improving estimation efficiency. We propose an estimator that is both doubly robust and semiparametrically efficient. Additionally, our method allows for a rigorous quantification of the efficiency gain through closed-form expressions. In general, ACPs are typically generated by various machine learning models, including but not limited to NLP and LLM. These predictions often come with measures of uncertainty quantification. An intriguing direction for future research is investigating how to effectively integrate these uncertainty quantification measures associated with ACPs into the learning process.

## Acknowledgement

We gratefully appreciate the ICML anonymous reviewers for their valuable feedback. The research is supported in part by NSF (DMS 1953526, 2122074, 2310942), NIH (R01DC021431) and the American Family Funding Initiative of UW-Madison.

## Impact Statement

This paper presents work whose goal is to advance the field of Machine Learning. Our study does not involve human subjects, complies with all legal and ethical standards, and we do not anticipate any potential harmful consequences resulting from our work.

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

# A. Technical Proofs.

*Proof of Proposition 3.1.* In the situation with incorporating ACPs, the i.i.d. data are $\{r_i = 1, y_i, \mathbf{x}_i, \widehat{y}_i\} \cup \{r_i = 0, \mathbf{x}_i, \widehat{y}_i\}$, $i = 1, \ldots, M$, and the likelihood function of one generic observation is

$$
\{p(y \mid \mathbf{x}, \widehat{y})p(\widehat{y} \mid \mathbf{x})p(\mathbf{x})\pi\}^r \{p(\widehat{y} \mid \mathbf{x})q(\mathbf{x})(1 - \pi)\}^{1-r}
$$
$$
= \quad p(y \mid \mathbf{x}, \widehat{y})^r p(\widehat{y} \mid \mathbf{x})p(\mathbf{x})^r q(\mathbf{x})^{1-r}\pi^r(1 - \pi)^{1-r}.
$$

We then consider the Hilbert space $\mathcal{H}$ of all $d$-dimensional zero-mean measurable functions with finite variance, equipped with the inner product $\langle \mathbf{h}_1, \mathbf{h}_2 \rangle = \mathrm{E}\{\mathbf{h}_1(\cdot)^\mathrm{T}\mathbf{h}_2(\cdot)\}$ where $\mathbf{h}_1(\cdot), \mathbf{h}_2(\cdot) \in \mathcal{H}$. We first give an orthogonal decomposition of the semiparametric tangent space $\mathcal{T}$ (Bickel et al., 1993; Tsiatis, 2006) that is defined as the mean squared closure of the tangent spaces of parametric submodels spanned by the score vectors. That is,

$$
\mathcal{T} = \Lambda_\pi \oplus \Lambda_p \oplus \Lambda_q \oplus \Lambda_{\widehat{y}} \oplus \Lambda_y,
$$

where

$$
\begin{aligned}
\Lambda_\pi &= \left\{ \left( \frac{r}{\pi} - \frac{1-r}{1-\pi} \right) \mathbf{a} : \forall \mathbf{a} \right\}, \\
\Lambda_p &= [r\mathbf{b}(\mathbf{x}) : \mathrm{E}_p\{\mathbf{b}(\mathbf{X})\} = \mathbf{0}], \\
\Lambda_q &= [(1-r)\mathbf{c}(\mathbf{x}) : \mathrm{E}_q\{\mathbf{c}(\mathbf{X})\} = \mathbf{0}], \\
\Lambda_{\widehat{y}} &= \left[\mathbf{d}(\widehat{y}, \mathbf{x}) : \mathrm{E}\{\mathbf{d}(\widehat{Y}, \mathbf{x}) \mid \mathbf{x}\} = \mathbf{0}\right], \\
\Lambda_y &= [r\mathbf{e}(y, \mathbf{x}, \widehat{y}) : \mathrm{E}\{\mathbf{e}(Y, \mathbf{x}, \widehat{y}) \mid \mathbf{x}, \widehat{y}\} = \mathbf{0}],
\end{aligned}
$$

are the tangent spaces with respect to $\pi$, $p(\mathbf{x})$, $q(\mathbf{x})$, $p(\widehat{y} \mid \mathbf{x})$ and $p(y \mid \mathbf{x}, \widehat{y})$, respectively. The notation $\oplus$ represents the direct sum of two spaces that are orthogonal to each other.

Recognizing that the EIF for estimating $\boldsymbol{\beta}$ is a special element in $\mathcal{T}$, we can assume it has the form

$$
\underbrace{r\mathbf{e}_1(y, \mathbf{x}, \widehat{y})}_{\in \Lambda_y} + \underbrace{\mathbf{d}_1(\widehat{y}, \mathbf{x})}_{\in \Lambda_{\widehat{y}}} + \underbrace{(1-r)\mathbf{c}_1(\mathbf{x})}_{\in \Lambda_q}.
$$

Define the score vectors as

$$
\begin{aligned}
\mathbf{s}_1(y, \mathbf{x}, \widehat{y}; \boldsymbol{\alpha}_1) &= \frac{\partial \log p(y \mid \mathbf{x}, \widehat{y}; \boldsymbol{\alpha}_1)}{\partial \boldsymbol{\alpha}_1}, \\
\mathbf{s}_2(\widehat{y}, \mathbf{x}; \boldsymbol{\alpha}_2) &= \frac{\partial \log p(\widehat{y} \mid \mathbf{x}; \boldsymbol{\alpha}_2)}{\partial \boldsymbol{\alpha}_2}, \\
\mathbf{s}_3(\mathbf{x}; \boldsymbol{\alpha}_3) &= \frac{\partial \log q(\mathbf{x}; \boldsymbol{\alpha}_3)}{\partial \boldsymbol{\alpha}_3}.
\end{aligned}
$$

Then, based on the orthogonality satisfied by the EIF (Theorem 4.2 and Theorem 4.3 in Tsiatis (2006)), we have

- From $\mathrm{E}(\mathbf{s}\mathbf{s}_1^\mathrm{T} \mid R = 0) = \mathrm{E}(R\mathbf{e}_1\mathbf{s}_1^\mathrm{T})$, one can derive that $\mathbf{e}_1 = \boldsymbol{\Omega}\left[\frac{1}{\pi}w_0(\mathbf{x})\{\mathbf{s}(y, \mathbf{x}; \boldsymbol{\beta}_0) - \widetilde{\mathbf{m}}_0(\mathbf{x}, \widehat{y})\}\right]$;

- From $\mathrm{E}(\mathbf{s}\mathbf{s}_2^\mathrm{T} \mid R = 0) = \mathrm{E}(\mathbf{d}_1\mathbf{s}_2^\mathrm{T}) = \mathrm{E}\left\{\frac{1-\pi}{1-\pi(\mathbf{x})}\mathbf{d}_1\mathbf{s}_2^\mathrm{T} \mid R = 0\right\}$, one can derive that

$$
\mathbf{d}_1 \quad = \quad \boldsymbol{\Omega}\left[\frac{1 - \pi_0(\mathbf{x})}{1 - \pi}\{\widetilde{\mathbf{m}}_0(\mathbf{x}, \widehat{y}) - \mathbf{m}_0(\mathbf{x})\}\right];
$$

- From $\mathrm{E}(\mathbf{s}\mathbf{s}_3^\mathrm{T} \mid R = 0) = \mathrm{E}\{(1 - R)\mathbf{c}_1\mathbf{s}_3^\mathrm{T}\}$, one can derive $\mathbf{c}_1 = \boldsymbol{\Omega}\left\{\frac{1}{1-\pi}\mathbf{m}_0(\mathbf{x})\right\}$,

and this completes the proof. $\quad\square$

*Proof of Proposition 3.3.* The asymptotic variance without ACP is

$$
\begin{aligned}
\boldsymbol{\Omega}\mathrm{E}(\boldsymbol{\phi}_{wo}^{\otimes 2})\boldsymbol{\Omega} &= \frac{1}{\pi}\mathrm{E}_q\left[w_0(\mathbf{X})\boldsymbol{\Omega}\{\mathbf{s}(Y,\mathbf{X};\boldsymbol{\beta}_0) - \mathbf{m}_0(\mathbf{X})\}^{\otimes 2}\boldsymbol{\Omega}\right] + \frac{1}{1-\pi}\mathrm{E}_q\left\{\boldsymbol{\Omega}\mathbf{m}_0(\mathbf{X})^{\otimes 2}\boldsymbol{\Omega}\right\}\\
&= \frac{1}{\pi}\mathrm{E}_q\left[w_0(\mathbf{X})\boldsymbol{\Omega}\{\mathbf{s}(Y,\mathbf{X};\boldsymbol{\beta}_0) - \widetilde{\mathbf{m}}_0(\mathbf{X},\widehat{Y})\}^{\otimes 2}\boldsymbol{\Omega}\right] + \frac{1}{\pi}\mathrm{E}_q\left[w_0(\mathbf{X})\boldsymbol{\Omega}\{\widetilde{\mathbf{m}}_0(\mathbf{X},\widehat{Y}) - \mathbf{m}_0(\mathbf{X})\}^{\otimes 2}\boldsymbol{\Omega}\right]\\
&\quad + \frac{1}{1-\pi}\mathrm{E}_q\left\{\boldsymbol{\Omega}\mathbf{m}_0(\mathbf{X})^{\otimes 2}\boldsymbol{\Omega}\right\}\\
&= \frac{1}{\pi}\mathrm{E}_q\left[w_0(\mathbf{X})\boldsymbol{\Omega}\{\mathbf{s}(Y,\mathbf{X};\boldsymbol{\beta}_0) - \widetilde{\mathbf{m}}_0(\mathbf{X},\widehat{Y})\}^{\otimes 2}\boldsymbol{\Omega}\right] + \frac{1}{1-\pi}\mathrm{E}_q\left\{\boldsymbol{\Omega}\mathbf{m}_0(\mathbf{X})^{\otimes 2}\boldsymbol{\Omega}\right\}\\
&\quad + \mathrm{E}\left[\frac{1}{\pi_0(\mathbf{X})}\frac{\{1-\pi_0(\mathbf{X})\}^2}{(1-\pi)^2}\boldsymbol{\Omega}\{\widetilde{\mathbf{m}}_0(\mathbf{X},\widehat{Y}) - \mathbf{m}_0(\mathbf{X})\}^{\otimes 2}\boldsymbol{\Omega}\right]
\end{aligned}
$$

The third equation uses $w_0(\mathbf{x}) = \frac{\pi}{1-\pi}\frac{1-\pi_0(\mathbf{x})}{\pi_0(\mathbf{x})}$.

The asymptotic variance with ACP is

$$
\begin{aligned}
\boldsymbol{\Omega}\mathrm{E}(\boldsymbol{\phi}_{w}^{\otimes 2})\boldsymbol{\Omega} &= \frac{1}{\pi}\mathrm{E}_q\left[w_0(\mathbf{X})\boldsymbol{\Omega}\{\mathbf{s}(Y,\mathbf{X};\boldsymbol{\beta}_0) - \widetilde{\mathbf{m}}_0(\mathbf{X},\widehat{Y})\}^{\otimes 2}\boldsymbol{\Omega}\right] + \frac{1}{1-\pi}\mathrm{E}_q\left\{\boldsymbol{\Omega}\mathbf{m}_0(\mathbf{X})^{\otimes 2}\boldsymbol{\Omega}\right\}\\
&\quad + \mathrm{E}\left[\left\{\frac{w_0(\mathbf{X})}{\pi + (1-\pi)w_0(\mathbf{X})}\right\}^2\boldsymbol{\Omega}\{\widetilde{\mathbf{m}}_0(\mathbf{X},\widehat{Y}) - \mathbf{m}_0(\mathbf{X})\}^{\otimes 2}\boldsymbol{\Omega}\right]\\
&= \frac{1}{\pi}\mathrm{E}_q\left[w_0(\mathbf{X})\boldsymbol{\Omega}\{\mathbf{s}(Y,\mathbf{X};\boldsymbol{\beta}_0) - \widetilde{\mathbf{m}}_0(\mathbf{X},\widehat{Y})\}^{\otimes 2}\boldsymbol{\Omega}\right] + \frac{1}{1-\pi}\mathrm{E}_q\left\{\boldsymbol{\Omega}\mathbf{m}_0(\mathbf{X})^{\otimes 2}\boldsymbol{\Omega}\right\}\\
&\quad + \mathrm{E}\left[\left\{\frac{1-\pi_0(\mathbf{X})}{(1-\pi)}\right\}^2\boldsymbol{\Omega}\{\widetilde{\mathbf{m}}_0(\mathbf{X},\widehat{Y}) - \mathbf{m}_0(\mathbf{X})\}^{\otimes 2}\boldsymbol{\Omega}\right]
\end{aligned}
$$

The second equation uses $\frac{1-\pi_0(\mathbf{x})}{1-\pi} = \frac{w_0(\mathbf{x})}{\pi+(1-\pi)w_0(\mathbf{x})}$.

Therefore, we have

$$
\boldsymbol{\Omega}\{\mathrm{E}(\boldsymbol{\phi}_{wo}^{\otimes 2}) - \mathrm{E}(\boldsymbol{\phi}_{w}^{\otimes 2})\}\boldsymbol{\Omega} = \frac{1}{(1-\pi)^2}\boldsymbol{\Omega}\mathrm{E}\left[\frac{\{1-\pi_0(\mathbf{X})\}^3}{\pi_0(\mathbf{X})}\{\widetilde{\mathbf{m}}_0(\mathbf{X},\widehat{Y}) - \mathbf{m}_0(\mathbf{X})\}^{\otimes 2}\right]\boldsymbol{\Omega}.
$$

$\square$

*Proof of Theorem 5.3.* When $w^*(\mathbf{x})$ is misspecified, and $\widetilde{\mathbf{m}}^*(\mathbf{x},\widehat{y})$ and $\mathbf{m}^*(\mathbf{x})$ are correctly specified. We have

$$
\begin{aligned}
&\mathrm{E}\left[\frac{R}{\pi}w^*(\mathbf{X})\{\mathbf{s}(Y,\mathbf{X};\boldsymbol{\beta}_0) - \widetilde{\mathbf{m}}^*(\mathbf{X},\widehat{Y})\} + \frac{w^*(\mathbf{X})}{(1-\pi)w^*(\mathbf{X}) + \pi}\{\widetilde{\mathbf{m}}^*(\mathbf{X},\widehat{Y}) - \mathbf{m}^*(\mathbf{X})\} + \frac{1-R}{1-\pi}\mathbf{m}^*(\mathbf{X})\right]\\
&= \mathrm{E}\left[\frac{R}{\pi}w^*(\mathbf{X})\{\mathbf{s}(Y,\mathbf{X};\boldsymbol{\beta}_0) - \widetilde{\mathbf{m}}_0(\mathbf{X},\widehat{Y})\} + \frac{w^*(\mathbf{X})}{(1-\pi)w^*(\mathbf{X}) + \pi}\{\widetilde{\mathbf{m}}_0(\mathbf{X},\widehat{Y}) - \mathbf{m}_0(\mathbf{X})\} + \frac{1-R}{1-\pi}\mathbf{m}_0(\mathbf{X})\right] = \mathbf{0}
\end{aligned}
$$

When $w^*(\mathbf{x})$ is correctly specified, and $\widetilde{\mathbf{m}}^*(\mathbf{x},\widehat{y})$ and $\mathbf{m}^*(\mathbf{x})$ are misspecified. We have

$$
\begin{aligned}
&\mathrm{E}\left[\frac{R}{\pi}w^*(\mathbf{X})\{\mathbf{s}(Y,\mathbf{X};\boldsymbol{\beta}_0) - \widetilde{\mathbf{m}}^*(\mathbf{X},\widehat{Y})\} + \frac{w^*(\mathbf{X})}{(1-\pi)w^*(\mathbf{X}) + \pi}\{\widetilde{\mathbf{m}}^*(\mathbf{X},\widehat{Y}) - \mathbf{m}^*(\mathbf{X})\} + \frac{1-R}{1-\pi}\mathbf{m}^*(\mathbf{X})\right]\\
&= \mathrm{E}\left[\frac{R}{\pi}w_0(\mathbf{X})\{\mathbf{s}(Y,\mathbf{X};\boldsymbol{\beta}_0) - \widetilde{\mathbf{m}}^*(\mathbf{X},\widehat{Y})\}\right] + \mathrm{E}\left[\frac{w_0(\mathbf{X})}{(1-\pi)w_0(\mathbf{X}) + \pi}\{\widetilde{\mathbf{m}}^*(\mathbf{X},\widehat{Y}) - \mathbf{m}^*(\mathbf{X})\}\right] + \mathrm{E}\left[\frac{1-R}{1-\pi}\mathbf{m}^*(\mathbf{X})\right]\\
&= \mathrm{E}\left[\frac{R}{\pi}w_0(\mathbf{X})\{\mathbf{s}(Y,\mathbf{X};\boldsymbol{\beta}_0) - \widetilde{\mathbf{m}}^*(\mathbf{X},\widehat{Y})\}\right] + \mathrm{E}\left[\frac{1-R}{1-\pi}\{\widetilde{\mathbf{m}}^*(\mathbf{X},\widehat{Y}) - \mathbf{m}^*(\mathbf{X})\}\right] + \mathrm{E}\left[\frac{1-R}{1-\pi}\mathbf{m}^*(\mathbf{X})\right]\\
&= \mathrm{E}\left[\frac{R}{\pi}w_0(\mathbf{X})\{\mathbf{s}(Y,\mathbf{X};\boldsymbol{\beta}_0) - \widetilde{\mathbf{m}}^*(\mathbf{X},\widehat{Y})\}\right] + \mathrm{E}\left[\frac{1-R}{1-\pi}\widetilde{\mathbf{m}}^*(\mathbf{X},\widehat{Y})\right] = \mathbf{0}.
\end{aligned}
$$

The proof is completed. $\square$

*Proof of Theorem 5.5.* The efficient influence function is

$$\phi_w(\rho_0, \boldsymbol{\beta}_0) = \frac{r}{\pi} w_0(\mathbf{x})\{\mathbf{s}(y, \mathbf{x}; \boldsymbol{\beta}_0) - \tilde{\mathbf{m}}_0(\mathbf{x}, \widehat{y})\} + \frac{w_0(\mathbf{x})}{(1 - \pi)w_0(\mathbf{x}) + \pi}\{\tilde{\mathbf{m}}_0(\mathbf{x}, \widehat{y}) - \mathbf{m}_0(\mathbf{x})\} + \frac{1 - r}{1 - \pi}\mathbf{m}_0(\mathbf{x}).$$

Then we have

$$\phi_w(\widehat{\rho}_k, \boldsymbol{\beta}_0) = \frac{r}{\pi} \widehat{w}_k(\mathbf{x})\{\mathbf{s}(y, \mathbf{x}; \boldsymbol{\beta}_0) - \widehat{\tilde{\mathbf{m}}}_k(\mathbf{x}, \widehat{y})\} + \frac{\widehat{w}_k(\mathbf{x})}{(1 - \pi)\widehat{w}_k(\mathbf{x}) + \pi}\{\widehat{\tilde{\mathbf{m}}}_k(\mathbf{x}, \widehat{y}) - \widehat{\mathbf{m}}_k(\mathbf{x})\} + \frac{1 - r}{1 - \pi}\widehat{\mathbf{m}}_k(\mathbf{x}).$$

Therefore,

$$\frac{1}{K}\sum_{k=1}^{K}\widehat{\mathrm{E}}_k\phi_w(\widehat{\rho}_k, \boldsymbol{\beta}_0) - \mathrm{E}\phi_w(\rho_0, \boldsymbol{\beta}_0) = \frac{1}{K}\sum_{k=1}^{K}\{\widehat{\mathrm{E}}_k\phi_1(\widehat{\rho}_k, \boldsymbol{\beta}_0) - \mathrm{E}\phi_1(\rho_0, \boldsymbol{\beta}_0)\}.$$

We only consider the following term,

$$\widehat{\mathrm{E}}_k\phi_w(\widehat{\rho}_k, \boldsymbol{\beta}_0) - \mathrm{E}\phi_w(\rho_0, \boldsymbol{\beta}_0)$$
$$= \underbrace{(\widehat{\mathrm{E}}_k - \mathrm{E})\{\phi_w(\widehat{\rho}_k, \boldsymbol{\beta}_0) - \phi_w(\rho_0, \boldsymbol{\beta}_0)\}}_{\text{sample splitting}} + \underbrace{\mathrm{E}\{\phi_w(\widehat{\rho}_k, \boldsymbol{\beta}_0) - \phi_w(\rho_0, \boldsymbol{\beta}_0)\}}_{\text{bias}} + \underbrace{(\widehat{\mathrm{E}}_k - \mathrm{E})\phi_w(\rho_0, \boldsymbol{\beta}_0)}_{\text{CLT}}.$$

(i) $(\widehat{\mathrm{E}}_k - \mathrm{E})\{\phi_w(\widehat{\rho}, \boldsymbol{\beta}_0) - \phi_w(\rho_0, \boldsymbol{\beta}_0)\} = o_p(M^{-1/2})$.

*Proof.* Let $\phi_w(\widehat{\rho}_k, \boldsymbol{\beta}_0)$ be a function estimated from the sample $D_k^c$, and let $\widehat{\mathrm{E}}_k$ denote the empirical measure over $D_k$. First note that, conditional on $D_k^c$, the term in question has mean zero since

$$\mathrm{E}[\widehat{\mathrm{E}}_k\{\phi_w(\widehat{\rho}_k, \boldsymbol{\beta}_0) - \phi_w(\rho_0, \boldsymbol{\beta}_0)\}|D_k^c] = \mathrm{E}\{\phi_w(\widehat{\rho}_k, \boldsymbol{\beta}_0) - \phi_w(\rho_0, \boldsymbol{\beta}_0)|D_k^c\}.$$

The conditional variance is

$$\mathrm{var}\left[(\widehat{\mathrm{E}}_k - \mathrm{E})\{\phi_w(\widehat{\rho}_k, \boldsymbol{\beta}_0) - \phi_w(\rho_0, \boldsymbol{\beta}_0)\}|D_k^c\right] = \mathrm{var}\left[\widehat{\mathrm{E}}_k\{\phi_w(\widehat{\rho}_k, \boldsymbol{\beta}_0) - \phi_w(\rho_0, \boldsymbol{\beta}_0)\}|D_k^c\right]$$
$$= \frac{2}{M}\mathrm{var}\{\phi_w(\widehat{\rho}_k, \boldsymbol{\beta}_0) - \phi_w(\rho_0, \boldsymbol{\beta}_0)|D_k^c\} \le 2\|\phi_w(\widehat{\rho}_k, \boldsymbol{\beta}_0) - \phi_w(\rho_0, \boldsymbol{\beta}_0)\|_2^2/M$$

Therefore using Chebyshev's inequality we have

$$\mathrm{pr}\left\{\frac{\|(\widehat{\mathrm{E}}_k - \mathrm{E})\{\phi_w(\widehat{\rho}_k, \boldsymbol{\beta}_0) - \phi_w(\rho_0, \boldsymbol{\beta}_0)\}\|_2}{\sqrt{2}\|\phi_w(\widehat{\rho}_k, \boldsymbol{\beta}_0) - \phi_w(\rho_0, \boldsymbol{\beta}_0)\|_2/\sqrt{M}} \ge t\right\}$$
$$= \mathrm{E}\left[\mathrm{pr}\left\{\frac{\|(\widehat{\mathrm{E}}_k - \mathrm{E})\{\phi_w(\widehat{\rho}_k, \boldsymbol{\beta}_0) - \phi_w(\rho_0, \boldsymbol{\beta}_0)\}\|_2}{\sqrt{2}\|\phi_w(\widehat{\rho}_k, \boldsymbol{\beta}_0) - \phi_w(\rho_0, \boldsymbol{\beta}_0)\|_2/\sqrt{M}} \ge t|D_k^c\right\}\right] \le \frac{1}{t^2}.$$

Thus for any $\varepsilon > 0$ we can pick $t = 1/\sqrt{\varepsilon}$ so that the probability above is no more than $\varepsilon$, which yields the result. $\square$

(ii) $\mathrm{E}\{\phi_w(\widehat{\rho}_k, \boldsymbol{\beta}_0) - \phi_w(\rho_0, \boldsymbol{\beta}_0)|D_k^c\} = O_p(\|\widehat{w}_k(\mathbf{x}) - w_0(\mathbf{x})\|_2\|\widehat{\tilde{\mathbf{m}}}_k(\mathbf{x}, \widehat{y}) - \tilde{\mathbf{m}}_0(\mathbf{x}, \widehat{y})\|_2) + O_p(\|\widehat{w}_k(\mathbf{x}) - w_0(\mathbf{x})\|_2\|\widehat{\mathbf{m}}_k(\mathbf{x}) - \mathbf{m}_0(\mathbf{x})\|_2) = o_p(M^{-1/2})$ by rate conditions.

*Proof.*

$$\mathrm{E}\{\phi_w(\widehat{\rho}_k, \boldsymbol{\beta}_0) - \phi_w(\rho_0, \boldsymbol{\beta}_0)|D_k^c\}$$
$$= \mathrm{E}\left(\frac{R}{\pi}\left[\widehat{w}_k(\mathbf{X})\{\mathbf{s}(Y, \mathbf{X}; \boldsymbol{\beta}_0) - \widehat{\tilde{\mathbf{m}}}_k(\mathbf{X}, \widehat{Y})\} - w_0(\mathbf{X})\{\mathbf{s}(Y, \mathbf{X}; \boldsymbol{\beta}_0) - \tilde{\mathbf{m}}_0(\mathbf{X}, \widehat{Y})\}\right]|D_k^c\right)$$
$$+ \mathrm{E}\left[\frac{\widehat{w}_k(\mathbf{X})}{(1 - \pi)\widehat{w}_k(\mathbf{X}) + \pi}\{\widehat{\tilde{\mathbf{m}}}_k(\mathbf{X}, \widehat{Y}) - \widehat{\mathbf{m}}_k(\mathbf{X})\} - \frac{w_0(\mathbf{X})}{(1 - \pi)w_0(\mathbf{X}) + \pi}\{\tilde{\mathbf{m}}_0(\mathbf{X}, \widehat{Y}) - \mathbf{m}_0(\mathbf{X})\}|D_k^c\right]$$
$$+ \mathrm{E}\left\{\frac{1 - R}{1 - \pi}\widehat{\mathbf{m}}_k(\mathbf{X}) - \frac{1 - R}{1 - \pi}\mathbf{m}_0(\mathbf{X})|D_k^c\right\}$$
$$=: (\mathrm{I}) + (\mathrm{II}) + (\mathrm{III})$$

For the first term (I), we have

$$
\begin{aligned}
\text{(I)} &= \mathrm{E}\left[\frac{R}{\pi}\{\widehat{w}_k(\mathbf{X}) - w_0(\mathbf{X})\}\{\mathbf{s}(Y, \mathbf{X}; \boldsymbol{\beta}_0) - \widetilde{\mathbf{m}}_0(\mathbf{X}, \widehat{Y})\}|D_k^c\right] + \mathrm{E}\left[\frac{R}{\pi}w_0(\mathbf{X})\{\widetilde{\mathbf{m}}_0(\mathbf{X}, \widehat{Y}) - \widehat{\widetilde{\mathbf{m}}}_k(\mathbf{X}, \widehat{Y})\}|D_k^c\right] \\
&\quad + \mathrm{E}\left[\frac{R}{\pi}\{\widehat{w}_k(\mathbf{X}) - w_0(\mathbf{X})\}\{\widetilde{\mathbf{m}}_0(\mathbf{X}, \widehat{Y}) - \widehat{\widetilde{\mathbf{m}}}_k(\mathbf{X}, \widehat{Y})\}|D_k^c\right] \\
&= \mathrm{E}\left[\frac{R}{\pi}w_0(\mathbf{X})\{\widetilde{\mathbf{m}}_0(\mathbf{X}, \widehat{Y}) - \widehat{\widetilde{\mathbf{m}}}_k(\mathbf{X}, \widehat{Y})\}|D_k^c\right] + \mathrm{E}\left[\frac{R}{\pi}\{\widehat{w}_k(\mathbf{X}) - w_0(\mathbf{X})\}\{\widetilde{\mathbf{m}}_0(\mathbf{X}, \widehat{Y}) - \widehat{\widetilde{\mathbf{m}}}_k(\mathbf{X}, \widehat{Y})\}|D_k^c\right].
\end{aligned}
$$

For the second term (II), we have

$$
\begin{aligned}
\text{(II)} &= \mathrm{E}\left(\left\{\frac{\widehat{w}_k(\mathbf{X})}{(1-\pi)\widehat{w}_k(\mathbf{X}) + \pi} - \frac{w_0(\mathbf{X})}{(1-\pi)w_0(\mathbf{X}) + \pi}\right\}[\{\widehat{\widetilde{\mathbf{m}}}_k(\mathbf{X}, \widehat{Y}) - \widetilde{\mathbf{m}}_0(\mathbf{X}, \widehat{Y})\} - \{\widehat{\mathbf{m}}_k(\mathbf{X}) - \mathbf{m}_0(\mathbf{X})\}]|D_k^c\right) \\
&\quad + \mathrm{E}\left[\left\{\frac{\widehat{w}_k(\mathbf{X})}{(1-\pi)\widehat{w}_k(\mathbf{X}) + \pi} - \frac{w_0(\mathbf{X})}{(1-\pi)w_0(\mathbf{X}) + \pi}\right\}\{\widetilde{\mathbf{m}}_0(\mathbf{X}, \widehat{Y}) - \mathbf{m}_0(\mathbf{X})\}|D_k^c\right] \\
&\quad + \mathrm{E}\left(\frac{w_0(\mathbf{X})}{(1-\pi)w_0(\mathbf{X}) + \pi}[\{\widehat{\widetilde{\mathbf{m}}}_k(\mathbf{X}, \widehat{Y}) - \widetilde{\mathbf{m}}_0(\mathbf{X}, \widehat{Y})\} - \{\widehat{\mathbf{m}}_k(\mathbf{X}) - \mathbf{m}_0(\mathbf{X})\}]|D_k^c\right) \\
&= \mathrm{E}\left(\left\{\frac{\widehat{w}_k(\mathbf{X})}{(1-\pi)\widehat{w}_k(\mathbf{X}) + \pi} - \frac{w_0(\mathbf{X})}{(1-\pi)w_0(\mathbf{X}) + \pi}\right\}[\{\widehat{\widetilde{\mathbf{m}}}_k(\mathbf{X}, \widehat{Y}) - \widetilde{\mathbf{m}}_0(\mathbf{X}, \widehat{Y})\} - \{\widehat{\mathbf{m}}_k(\mathbf{X}) - \mathbf{m}_0(\mathbf{X})\}]|D_k^c\right) \\
&\quad + \mathrm{E}\left(\frac{w_0(\mathbf{X})}{(1-\pi)w_0(\mathbf{X}) + \pi}[\{\widehat{\widetilde{\mathbf{m}}}_k(\mathbf{X}, \widehat{Y}) - \widetilde{\mathbf{m}}_0(\mathbf{X}, \widehat{Y})\} - \{\widehat{\mathbf{m}}_k(\mathbf{X}) - \mathbf{m}_0(\mathbf{X})\}]|D_k^c\right).
\end{aligned}
$$

For the third term (III), we have

$$
\text{(III)} = \mathrm{E}\left[\frac{1-R}{1-\pi}\{\widehat{\mathbf{m}}_k(\mathbf{X}) - \mathbf{m}_0(\mathbf{X})\}|D_k^c\right]
$$

Combine (I), (II) and (III), we obtain

$$
\begin{aligned}
\text{(I)} + \text{(II)} + \text{(III)} &= O_p(\|\widehat{w}_k(\mathbf{x}) - w_0(\mathbf{x})\|_2\|\widehat{\widetilde{\mathbf{m}}}_k(\mathbf{x}, \widehat{y}) - \widetilde{\mathbf{m}}_0(\mathbf{x}, \widehat{y})\|_2) + O_p(\|\widehat{w}_k(\mathbf{x}) - w_0(\mathbf{x})\|_2\|\widehat{\mathbf{m}}_k(\mathbf{x}) - \mathbf{m}_0(\mathbf{x})\|_2) \\
&= o_p(M^{-1/2}).
\end{aligned}
$$

The second equation by the rate conditions. $\qquad\square$

Therefore, we have

$$
\widehat{\boldsymbol{\beta}}(\widehat{\rho}) - \boldsymbol{\beta}_0(\rho_0) = \widehat{\boldsymbol{\Omega}}\frac{1}{K}\sum_{k=1}^{K}\widehat{\mathrm{E}}_k\boldsymbol{\phi}_w(\widehat{\rho}_k, \boldsymbol{\beta}_0) - \boldsymbol{\Omega}\mathrm{E}\boldsymbol{\phi}_w(\rho_0, \boldsymbol{\beta}_0) = \widehat{\boldsymbol{\Omega}}\frac{1}{K}\sum_{k=1}^{K}\widehat{\mathrm{E}}_k\boldsymbol{\phi}_w(\widehat{\rho}_k, \boldsymbol{\beta}_0) - \widehat{\boldsymbol{\Omega}}\mathrm{E}\boldsymbol{\phi}_w(\rho_0, \boldsymbol{\beta}_0)
$$

$$
= \widehat{\boldsymbol{\Omega}}\frac{1}{K}\sum_{k=1}^{K}\underbrace{(\widehat{\mathrm{E}}_k - \mathrm{E})\{\boldsymbol{\phi}_w(\widehat{\rho}_k, \boldsymbol{\beta}_0) - \boldsymbol{\phi}_w(\rho_0, \boldsymbol{\beta}_0)\}}_{\text{sample splitting}} + \widehat{\boldsymbol{\Omega}}\frac{1}{K}\sum_{k=1}^{K}\underbrace{\mathrm{E}\{\boldsymbol{\phi}_w(\widehat{\rho}_k, \boldsymbol{\beta}) - \boldsymbol{\phi}_w(\rho_0, \boldsymbol{\beta}_0)\}}_{\text{bias}} + \widehat{\boldsymbol{\Omega}}\frac{1}{K}\sum_{k=1}^{K}\underbrace{(\widehat{\mathrm{E}}_k - \mathrm{E})\boldsymbol{\phi}_w(\rho_0, \boldsymbol{\beta}_0)}_{\text{CLT}}
$$

$$
= \boldsymbol{\Omega}\frac{1}{K}\sum_{k=1}^{K}\underbrace{(\widehat{\mathrm{E}}_k - \mathrm{E})\boldsymbol{\phi}_w(\rho_0, \boldsymbol{\beta}_0)}_{\text{CLT}} + o_p(M^{-1/2})
$$

Based on central limit theorem, We have

$$
\begin{aligned}
\sqrt{M}\{\widehat{\boldsymbol{\beta}}(\widehat{\rho}) - \boldsymbol{\beta}_0(\rho_0)\} &= \sqrt{M}\boldsymbol{\Omega}\frac{1}{K}\sum_{k=1}^{K}(\widehat{\mathrm{E}}_k - \mathrm{E})\boldsymbol{\phi}_w(\rho_0, \boldsymbol{\beta}_0) + o_p(1) \\
&\xrightarrow{d} N(\mathbf{0}, \boldsymbol{\Omega}\mathbf{V}_w\boldsymbol{\Omega}).
\end{aligned}
$$

The proof is completed. $\qquad\square$

*Proof of Theorem 5.8.* In this section, we divide it into two parts. First, we prove the consistency of the covariance matrix, and then prove the confidence interval.

First, we need to show the $\widehat{\boldsymbol{\Omega}}\widehat{\mathbf{V}}_w\widehat{\boldsymbol{\Omega}}$ is consistent. Here, we only show $\widehat{\mathbf{V}}_w = \frac{1}{K}\sum_{k=1}^{K}\widehat{\mathbb{E}}_k\{\phi_w^{\otimes 2}(\widehat{\rho}_k;\widehat{\boldsymbol{\beta}})\}$ is consistent. Note that

$$
\begin{aligned}
\|\widehat{\mathbf{V}}_w - \mathbf{V}_w\|_2 &= \|\widehat{\mathbf{V}}_w - \mathrm{E}\{\phi_w^{\otimes 2}(\rho_0;\boldsymbol{\beta}_0)\}\|_2 \\
&\leq \frac{1}{K}\sum_{k=1}^{K}\|\widehat{\mathbb{E}}_k\{\phi_w^{\otimes 2}(\widehat{\rho}_k;\widehat{\boldsymbol{\beta}})\} - \mathrm{E}\{\phi_w^{\otimes 2}(\rho_0;\boldsymbol{\beta}_0)\}\|_2.
\end{aligned}
$$

We only need to prove that $\|\widehat{\mathbb{E}}_k\{\phi_w^{\otimes 2}(\widehat{\rho}_k;\widehat{\boldsymbol{\beta}})\} - \mathrm{E}\{\phi_w^{\otimes 2}(\rho_0;\boldsymbol{\beta}_0)\}\|_2 = o_p(1)$. Consider the following decomposition:

$$
\begin{aligned}
&\|\widehat{\mathbb{E}}_k\{\phi_1^{\otimes 2}(\widehat{\rho}_k;\widehat{\boldsymbol{\beta}})\} - \mathrm{E}\{\phi_w^{\otimes 2}(\rho_0;\boldsymbol{\beta}_0)\}\|_2 \\
\leq\ & \|\widehat{\mathbb{E}}_k\{\phi_w^{\otimes 2}(\widehat{\rho}_k;\widehat{\boldsymbol{\beta}})\} - \widehat{\mathbb{E}}_k\{\phi_w^{\otimes 2}(\rho_0;\boldsymbol{\beta}_0)\}\|_2 + \|\widehat{\mathbb{E}}_k\{\phi_w^{\otimes 2}(\rho_0;\boldsymbol{\beta}_0)\} - \mathrm{E}\{\phi_w^{\otimes 2}(\rho_0;\boldsymbol{\beta}_0)\}\|_2 \\
=\ & R_1 + R_2.
\end{aligned}
$$

Thus we only need to prove that both $R_1$ and $R_2$ are $o_p(1)$.

By the law of large numbers, we can easily obtain $R_2 = o_p(1)$. Next we analyze the term $R_1$.

$$
\begin{aligned}
&\|\widehat{\mathbb{E}}_k\{\phi_w^{\otimes 2}(\widehat{\rho}_k;\widehat{\boldsymbol{\beta}})\} - \widehat{\mathbb{E}}_k\{\phi_w^{\otimes 2}(\rho_0;\boldsymbol{\beta}_0)\}\|_2 \\
\leq\ & \|\widehat{\mathbb{E}}_k\{\phi_w(\widehat{\rho}_k;\widehat{\boldsymbol{\beta}}) - \phi_w(\rho_0;\boldsymbol{\beta}_0)\}^{\otimes 2}\|_2 + 2\|\widehat{\mathbb{E}}_k\{\phi_w(\widehat{\rho}_k;\widehat{\boldsymbol{\beta}}) - \phi_w(\rho_0;\boldsymbol{\beta}_0)\}\{\phi_w(\rho_0;\boldsymbol{\beta}_0)\}^{\mathrm{T}}\|_2 \\
\leq\ & R_3^{1/2} \times \{R_3^{1/2} + 2\|\widehat{\mathbb{E}}_k\phi_w^{\otimes 2}(\rho_0;\boldsymbol{\beta}_0)\|_2^{1/2}\}
\end{aligned}
$$

where $R_3 = \|\widehat{\mathbb{E}}_k\{\phi_w(\widehat{\rho}_k;\widehat{\boldsymbol{\beta}}) - \phi_w(\rho_0;\boldsymbol{\beta}_0)\}^{\otimes 2}\|_2$.

Since $\mathrm{E}\phi_w^{\otimes 2}(\rho_0;\boldsymbol{\beta}_0) = O(1)$, Markov inequality implies that $\widehat{\mathbb{E}}_k\phi_w^{\otimes 2}(\rho_0;\boldsymbol{\beta}_0) = O_p(1)$. Moreover,

$$
R_3 \leq 2C_1\|\widehat{\boldsymbol{\beta}} - \boldsymbol{\beta}_0\|_2^2 + 2\|\widehat{\mathbb{E}}_k\{\phi_w(\widehat{\rho}_k;\boldsymbol{\beta}_0) - \phi_w(\rho_0;\boldsymbol{\beta}_0)\}^{\otimes 2}\|_2
$$

Since $\|\widehat{\boldsymbol{\beta}} - \boldsymbol{\beta}_0\|_2 = o_p(1)$ by Theorem 5.3, thus we only to prove $\|\widehat{\mathbb{E}}_k\{\phi_w(\widehat{\rho}_k;\boldsymbol{\beta}_0) - \phi_w(\rho_0;\boldsymbol{\beta}_0)\}^{\otimes 2}\|_2 = o_p(1)$ as well. We can further decompose this term:

$$
\begin{aligned}
&\widehat{\mathbb{E}}_k\{\phi_1(\widehat{\rho}_k;\boldsymbol{\beta}_0) - \phi_1(\rho_0;\boldsymbol{\beta}_0)\}^{\otimes 2} \\
=\ & \widehat{\mathbb{E}}_k\{\phi_1(\widehat{\rho}_k;\boldsymbol{\beta}_0) - \phi_1(\rho_0;\boldsymbol{\beta}_0)\}^{\otimes 2} - \mathrm{E}\{\phi_1(\widehat{\rho}_k;\boldsymbol{\beta}_0) - \phi_1(\rho_0;\boldsymbol{\beta}_0)\}^2 \\
& + \mathrm{E}\{\phi_1(\widehat{\rho}_k;\boldsymbol{\beta}_0) - \phi_1(\rho_0;\boldsymbol{\beta}_0)\}^{\otimes 2} = (1) + (2).
\end{aligned}
$$

Note that $\mathrm{E}\{(1)\} = \mathbf{0}$, so by Markov inequality, we have $(1) = o_p(1)$. Moreover, it is easy to verify that $(2) = o_p(1)$ as $O_p(\|\widehat{w}_k(\mathbf{x}) - w_0(\mathbf{x})\|_2\|\widehat{\widetilde{\mathbf{m}}}_k(\mathbf{x},\widehat{y}) - \widetilde{\mathbf{m}}_0(\mathbf{x},\widehat{y})\|_2) + O_p(\|\widehat{w}_k(\mathbf{x}) - w_0(\mathbf{x})\|_2\|\widehat{\mathbf{m}}_k(\mathbf{x}) - \mathbf{m}_0(\mathbf{x})\|_2) = o_p(1)$.

Putting all above together, we have

$$
\|R_1\|_2 = o_p(1).
$$

Thus, we have $\widehat{\mathbf{V}}_1 = \mathbf{V}_1 + o_p(1)$. Further, we have $\widehat{\mathbf{V}}_w = \mathbf{V}_w + o_p(1)$ by $\widehat{\boldsymbol{\Omega}} = \boldsymbol{\Omega} + o_p(1)$.

Second, we will show that $\mathbf{v}^{\mathrm{T}}\boldsymbol{\beta}_0 \in \mathrm{CI}_\alpha$ with probability $1 - \alpha$ in the limit; that is,

$$
\lim_{M\to\infty}\mathrm{pr}\left(\mathbf{v}^{\mathrm{T}}\boldsymbol{\beta}_0 \in \mathrm{CI}_\alpha\right) = 1 - \alpha.
$$

It can be easy to obtain, since we have

$$\lim_{M\to\infty} \mathrm{pr}\left(\mathbf{v}^{\mathrm{T}}\boldsymbol{\beta}_0 \in \mathrm{CI}_\alpha\right)$$

$$= \lim_{M\to\infty} \mathrm{pr}\left\{\mathbf{v}^{\mathrm{T}}\widehat{\boldsymbol{\beta}} - \Phi^{-1}(1-\alpha/2)\sqrt{\mathbf{v}^{\mathrm{T}}\widehat{\boldsymbol{\Omega}}\widehat{\mathbf{V}}_w\widehat{\boldsymbol{\Omega}}\mathbf{v}/M} \le \mathbf{v}^{\mathrm{T}}\boldsymbol{\beta}_0 \le \mathbf{v}^{\mathrm{T}}\widehat{\boldsymbol{\beta}} + \Phi^{-1}(1-\alpha/2)\sqrt{\mathbf{v}^{\mathrm{T}}\widehat{\boldsymbol{\Omega}}\widehat{\mathbf{V}}_w\widehat{\boldsymbol{\Omega}}\mathbf{v}/M}\right\}$$

$$= \lim_{M\to\infty} \mathrm{pr}\left\{-\Phi^{-1}(1-\alpha/2) \le \frac{\sqrt{M}(\mathbf{v}^{\mathrm{T}}\boldsymbol{\beta}_0 - \mathbf{v}^{\mathrm{T}}\widehat{\boldsymbol{\beta}})}{\sqrt{\mathbf{v}^{\mathrm{T}}\widehat{\boldsymbol{\Omega}}\widehat{\mathbf{V}}_w\widehat{\boldsymbol{\Omega}}\mathbf{v}}} \le \Phi^{-1}(1-\alpha/2)\right\}$$

$$= \lim_{M\to\infty} \mathrm{pr}\left\{\frac{\sqrt{M}(\mathbf{v}^{\mathrm{T}}\boldsymbol{\beta}_0 - \mathbf{v}^{\mathrm{T}}\widehat{\boldsymbol{\beta}})}{\sqrt{\mathbf{v}^{\mathrm{T}}\widehat{\boldsymbol{\Omega}}\widehat{\mathbf{V}}_w\widehat{\boldsymbol{\Omega}}\mathbf{v}}} \le \Phi^{-1}(1-\alpha/2)\right\} - \lim_{M\to\infty} \mathrm{pr}\left\{\frac{\sqrt{M}(\mathbf{v}^{\mathrm{T}}\boldsymbol{\beta}_0 - \mathbf{v}^{\mathrm{T}}\widehat{\boldsymbol{\beta}})}{\sqrt{\mathbf{v}^{\mathrm{T}}\widehat{\boldsymbol{\Omega}}\widehat{\mathbf{V}}_w\widehat{\boldsymbol{\Omega}}\mathbf{v}}} < -\Phi^{-1}(1-\alpha/2)\right\}$$

$$= \Phi\{\Phi^{-1}(1-\alpha/2)\} - \Phi\{-\Phi^{-1}(1-\alpha/2)\}$$

$$= 1 - \alpha/2 - \alpha/2 = 1 - \alpha$$

The fourth equation holds because $\sqrt{M}(\mathbf{v}^{\mathrm{T}}\boldsymbol{\beta}_0 - \mathbf{v}^{\mathrm{T}}\widehat{\boldsymbol{\beta}})/\sqrt{\mathbf{v}^{\mathrm{T}}\widehat{\boldsymbol{\Omega}}\widehat{\mathbf{V}}_w\widehat{\boldsymbol{\Omega}}\mathbf{v}} \xrightarrow{d} N(0,1)$. The fifth equation holds because the standard normal distribution is symmetric. The proof is completed. $\square$

## B. Parallel Results for Parameter $\theta$ in the Combined Population

To avoid repetition, we only present the results for the parameter $\boldsymbol{\beta}$, some characteristic in the unlabeled data population $\mathcal{U}$, in the main paper. But, all the results can be generalized to the characteristic of the combind data population $\mathcal{L} \cup \mathcal{U}$ if it is of interest.

Similarly, we define the $d$-dimensional parameter $\boldsymbol{\theta}$ as

$$\boldsymbol{\theta} = \mathrm{argmin}\, \mathrm{E}\{\ell(y, \mathbf{x}; \boldsymbol{\theta})\},$$

and it is equivalent to write $\boldsymbol{\theta}$ as the solution of the estimating equation

$$\mathrm{E}\{\mathbf{u}(Y, \mathbf{X}; \boldsymbol{\theta})\} = \mathbf{0}.$$

To proceed, we assume the $d \times d$ matrix $\mathrm{E}\{\partial\mathbf{u}(Y, \mathbf{X}; \boldsymbol{\theta})/\partial\boldsymbol{\theta}^{\mathrm{T}}\}$ evaluated at the true value $\boldsymbol{\theta}_0$ is invertible and denote the inverse as $\boldsymbol{\Gamma}$. We also denote $\mathrm{E}\{\mathbf{u}(Y, \mathbf{x}; \boldsymbol{\theta}) \mid \mathbf{x}\}$ as $\mathbf{h}(\mathbf{x})$ and denote $\mathrm{E}\{\mathbf{u}(Y, \mathbf{x}; \boldsymbol{\theta}) \mid \mathbf{x}, \widehat{y}\}$ as $\widetilde{\mathbf{h}}(\mathbf{x}, \widehat{y})$.

For estimating $\boldsymbol{\theta}$, in the situation with ACPs, the EIF equals $\boldsymbol{\Gamma}\boldsymbol{\varphi}_w$ with

$$\boldsymbol{\varphi}_w = \frac{r}{\pi_0(\mathbf{x})}\{\mathbf{u}(y, \mathbf{x}; \boldsymbol{\theta}_0) - \widetilde{\mathbf{h}}_0(\mathbf{x}, \widehat{y})\} + \widetilde{\mathbf{h}}_0(\mathbf{x}, \widehat{y}),$$

and the efficiency bound is $\boldsymbol{\Gamma}\mathbf{V}_w\boldsymbol{\Gamma}$ with

$$\mathbf{V}_w = \mathrm{E}\left[\frac{R}{\pi_0(\mathbf{X})^2}\{\mathbf{u}(Y, \mathbf{X}; \boldsymbol{\theta}_0) - \widetilde{\mathbf{h}}_0(\mathbf{X}, \widehat{Y})\}^{\otimes 2}\right] + \mathrm{E}\left[\{\widetilde{\mathbf{h}}_0(\mathbf{X}, \widehat{Y}) - \mathbf{h}_0(\mathbf{X})\}^{\otimes 2}\right] + \mathrm{E}\left\{\mathbf{h}_0(\mathbf{X})^{\otimes 2}\right\}.$$

In the situation without ACPs, the EIF is $\boldsymbol{\Gamma}\boldsymbol{\varphi}_{wo}$ with

$$\boldsymbol{\varphi}_{wo} = \frac{r}{\pi_0(\mathbf{x})}\{\mathbf{u}(y, \mathbf{x}; \boldsymbol{\theta}_0) - \mathbf{h}_0(\mathbf{x})\} + \mathbf{h}_0(\mathbf{x}),$$

and the efficiency bound is $\boldsymbol{\Gamma}\mathbf{V}_{wo}\boldsymbol{\Gamma}$ with

$$\mathbf{V}_{wo} = \mathrm{E}\left[\frac{R}{\pi_0(\mathbf{X})^2}\{\mathbf{u}(Y, \mathbf{X}; \boldsymbol{\theta}_0) - \widetilde{\mathbf{h}}_0(\mathbf{X}, \widehat{Y})\}^{\otimes 2}\right] + \mathrm{E}\left[\frac{R}{\pi_0(\mathbf{X})^2}\{\widetilde{\mathbf{h}}_0(\mathbf{X}, \widehat{Y}) - \mathbf{h}_0(\mathbf{X})\}^{\otimes 2}\right] + \mathrm{E}\left\{\mathbf{h}_0(\mathbf{X})^{\otimes 2}\right\}.$$

Therefore, one can compute that the efficiency gain of using ACP $\widehat{Y}$ is

$$\boldsymbol{\Gamma}(\mathbf{V}_{wo} - \mathbf{V}_w)\boldsymbol{\Gamma} = \boldsymbol{\Gamma}\mathrm{E}\left[\left\{\frac{R}{\pi_0(\mathbf{X})^2} - 1\right\}\{\widetilde{\mathbf{h}}_0(\mathbf{X}, \widehat{Y}) - \mathbf{h}_0(\mathbf{X})\}^{\otimes 2}\right]\boldsymbol{\Gamma}$$

$$= \boldsymbol{\Gamma}\mathrm{E}\left[\frac{1 - \pi_0(\mathbf{X})}{\pi_0(\mathbf{X})}\{\widetilde{\mathbf{h}}_0(\mathbf{X}, \widehat{Y}) - \mathbf{h}_0(\mathbf{X})\}^{\otimes 2}\right]\boldsymbol{\Gamma},$$

which is positive definite, as long as $\pi_0(\mathbf{x})$ is bounded away from zero and one and that the ACP $\widehat{Y}$ does depend on some other variable $\mathbf{Z}$, beyond the available feature $\mathbf{X}$ in both labeled and unlabeled data. Clearly, this whole rationale is the same as that for the parameter $\boldsymbol{\beta}$, as well as the following theoretical results including double robustness and semiparametric efficiency.

## C. Additional Numerical Results

In this section, we provide more numerical results for various parameters of interest across different models. For each parameter, we demonstrate the efficiency improvement by incorporating ACPs under varying conditions, including varying labeled sample size ($n$), varying unlabeled sample size ($N$), varying signal strength from ACP ($\alpha$), and varying correlation between ACP and the covariates ($\zeta$).

Figures 4-5 present the results for various parameters under the linear model, while Figures 6-8 illustrate the results for the logistic model. Similarly, Tables 5-8 and Tables 9-12 summarize the results for various parameters under the linear and logistic models, respectively.

In addition, we also present additional numerical results and comparisons with benchmark methods–PPI, PPI++, and RePPI–across various parameters of interest under different modeling scenarios. For each parameter, we evaluate the efficiency gains achieved by incorporating ACP under a range of conditions, including varying the labeled sample size ($n$), the unlabeled sample size ($N$), the signal strength of the ACP ($\alpha$), and the correlation between the ACP and covariates ($\zeta$). Table 13 reports the comparison results for estimating the outcome mean under linear models, while Table 14 presents the corresponding results for coefficient estimation under linear models.

| $n$ | Method | $\mathrm{MSE}(\overline{Y})$ | $\mathrm{MSE}(\widehat{\xi}_1)$ | $\mathrm{MSE}(\widehat{\xi}_2)$ | $\mathrm{MSE}(\widehat{\xi}_3)$ | $\mathrm{MSE}(\widehat{\xi}_4)$ | $\mathrm{MSE}(\widehat{\xi}_5)$ |
|------|---------|------|------|------|------|------|------|
| 300 | w.o. ACP | 0.84 | 0.82 | 0.86 | 1.00 | 0.93 | 0.93 |
| 300 | w. ACP | 0.08 | 0.10 | 0.08 | 0.10 | 0.09 | 0.10 |
| 600 | w.o. ACP | 0.83 | 0.80 | 0.80 | 0.97 | 0.92 | 0.90 |
| 600 | w. ACP | 0.06 | 0.06 | 0.06 | 0.07 | 0.06 | 0.06 |
| 900 | w.o. ACP | 0.82 | 0.80 | 0.79 | 0.95 | 0.93 | 0.90 |
| 900 | w. ACP | 0.05 | 0.05 | 0.05 | 0.06 | 0.05 | 0.06 |
| 1200 | w.o. ACP | 0.81 | 0.79 | 0.79 | 0.94 | 0.94 | 0.90 |
| 1200 | w. ACP | 0.05 | 0.05 | 0.04 | 0.05 | 0.05 | 0.05 |
| 1500 | w.o. ACP | 0.81 | 0.79 | 0.78 | 0.93 | 0.92 | 0.91 |
| 1500 | w. ACP | 0.05 | 0.04 | 0.04 | 0.05 | 0.04 | 0.04 |

*Table 5.* Simulation results under different $n$ when $\alpha = 5$, $\zeta = 0$, $N = 300$ under the linear model setting.

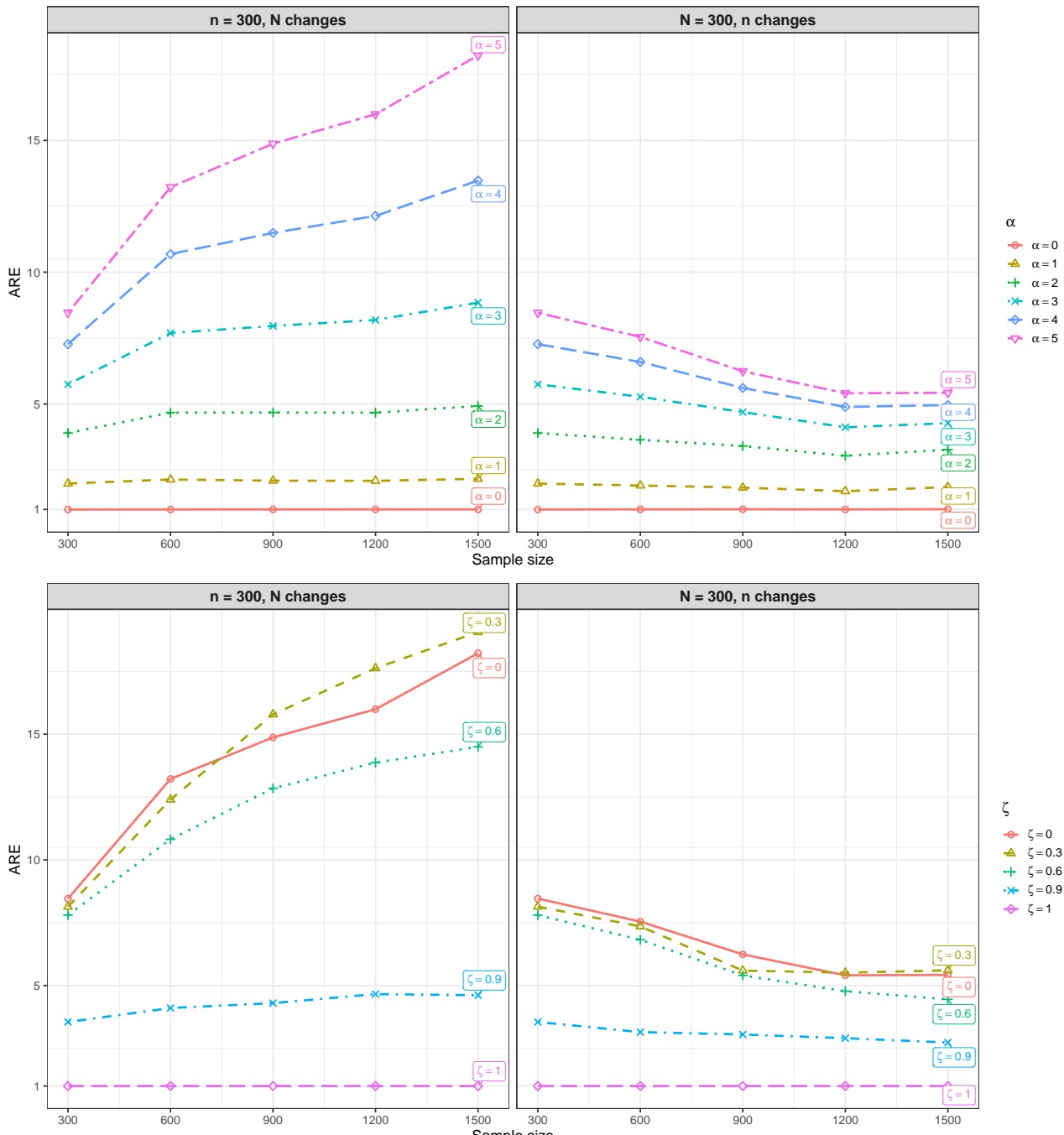

*Figure 4.* Variation of ARE for the estimation of $\xi_1$ under linear model setting with different sample sizes, signal strength, and correlation coefficient.

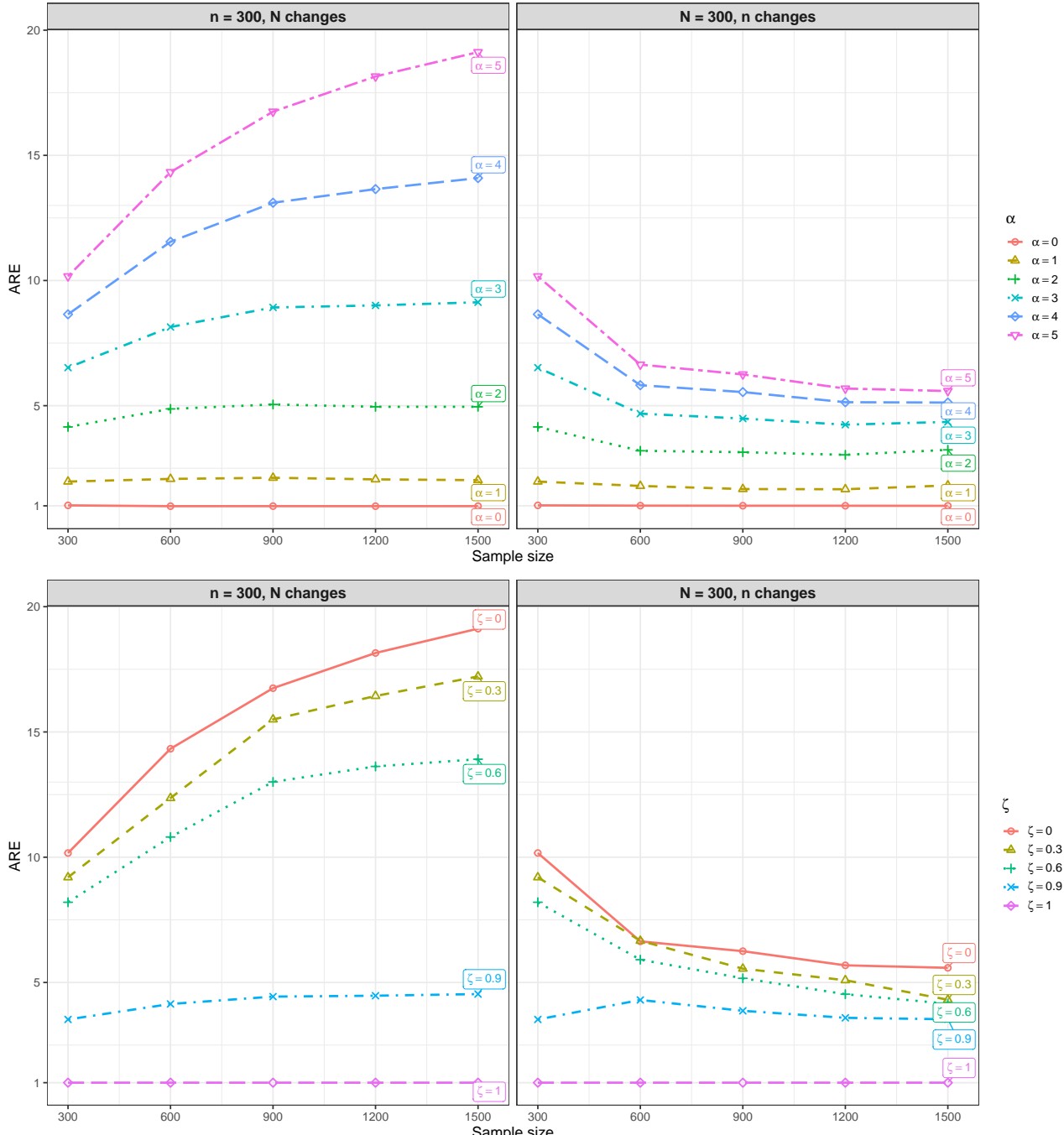

*Figure 5.* Variation of ARE for the estimation of $\xi_2$ under linear model setting with different sample sizes, signal strength, and correlation coefficient.

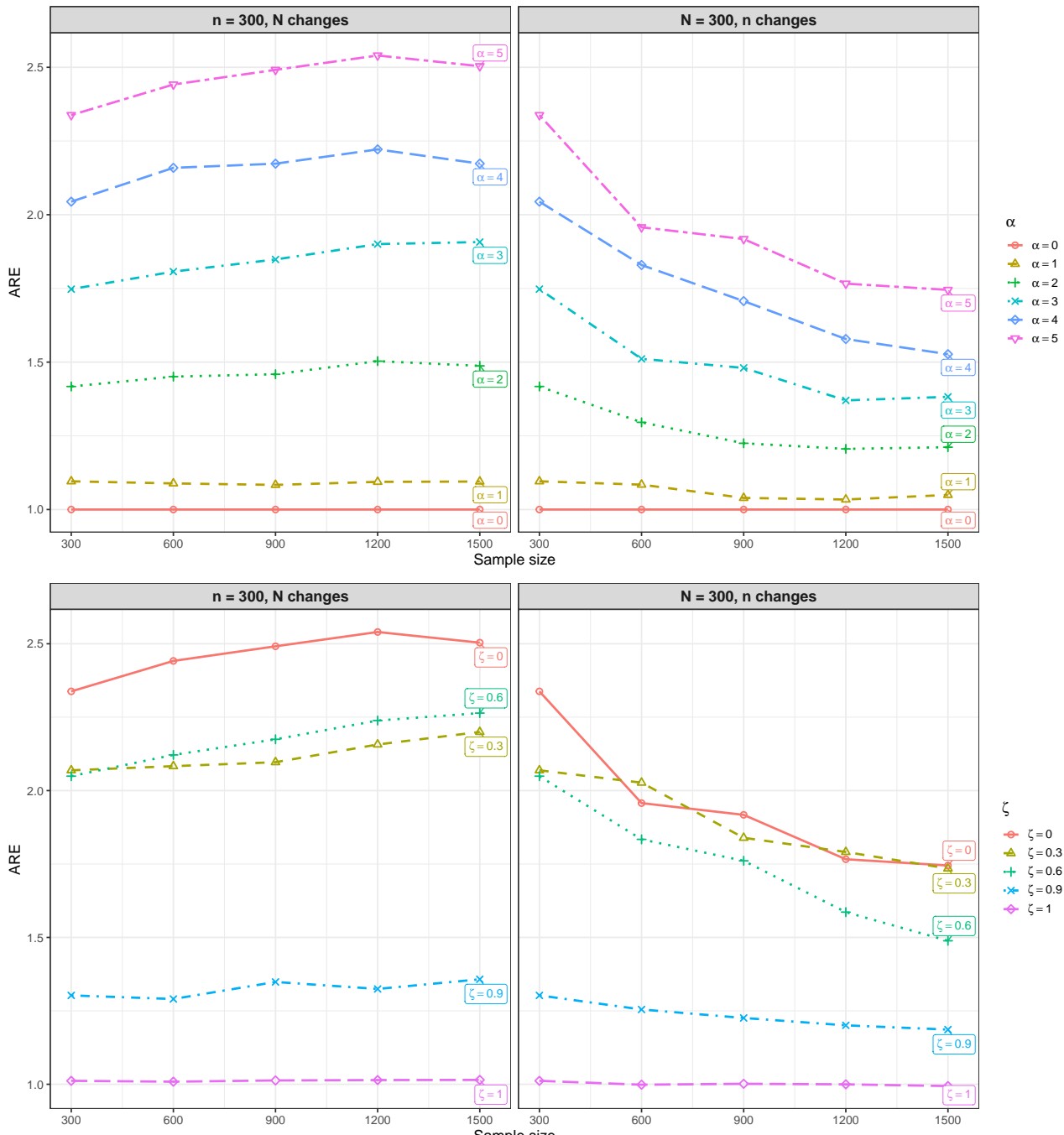

*Figure 6.* Variation of ARE for the estimation of $\mathrm{E}_q(Y)$ under logistic model setting with different sample sizes, signal strength, and correlation coefficient.

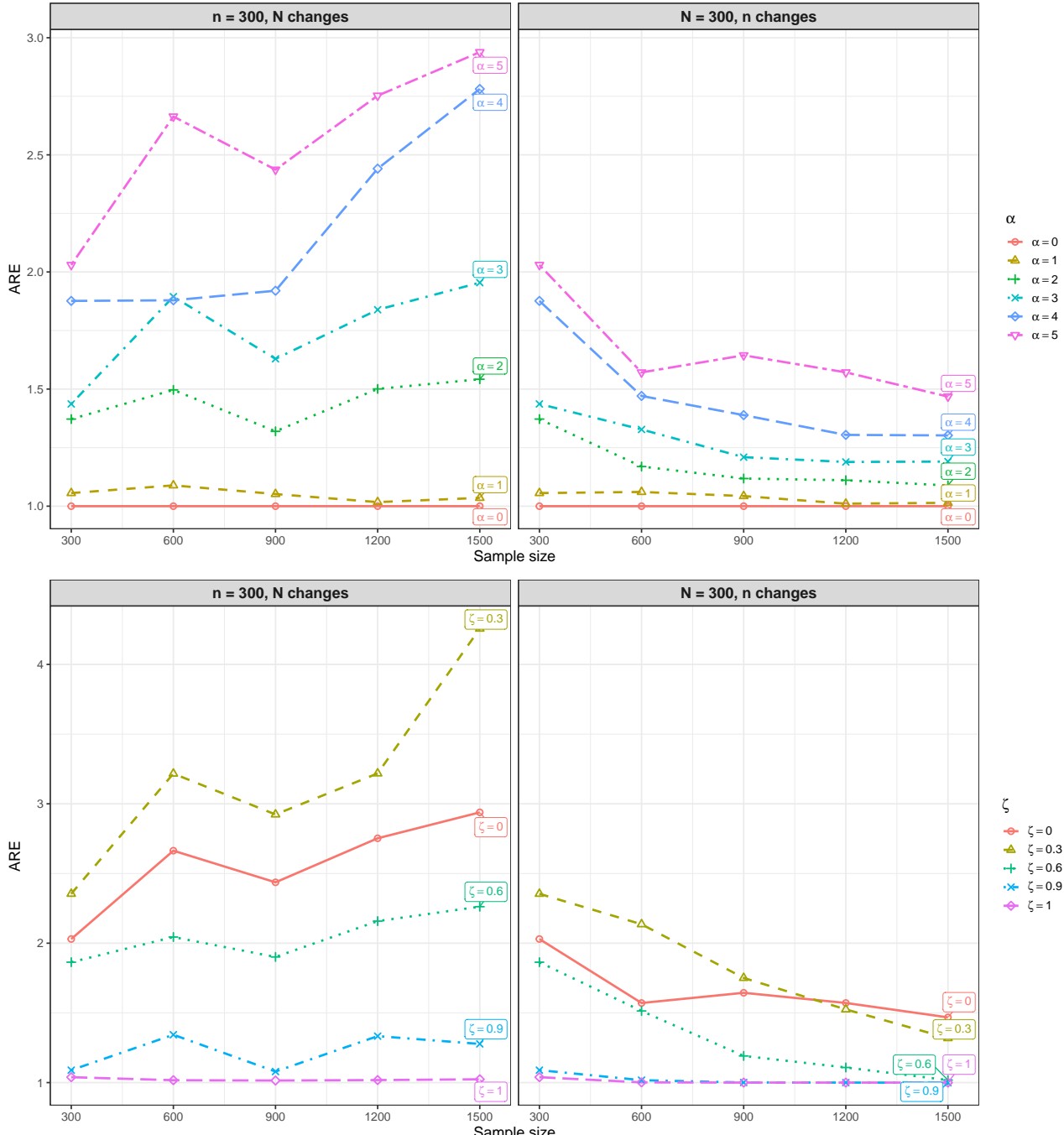

*Figure 7.* Variation of ARE for the estimation of $\xi_1$ under logistic model setting with different sample sizes, signal strength, and correlation coefficient.

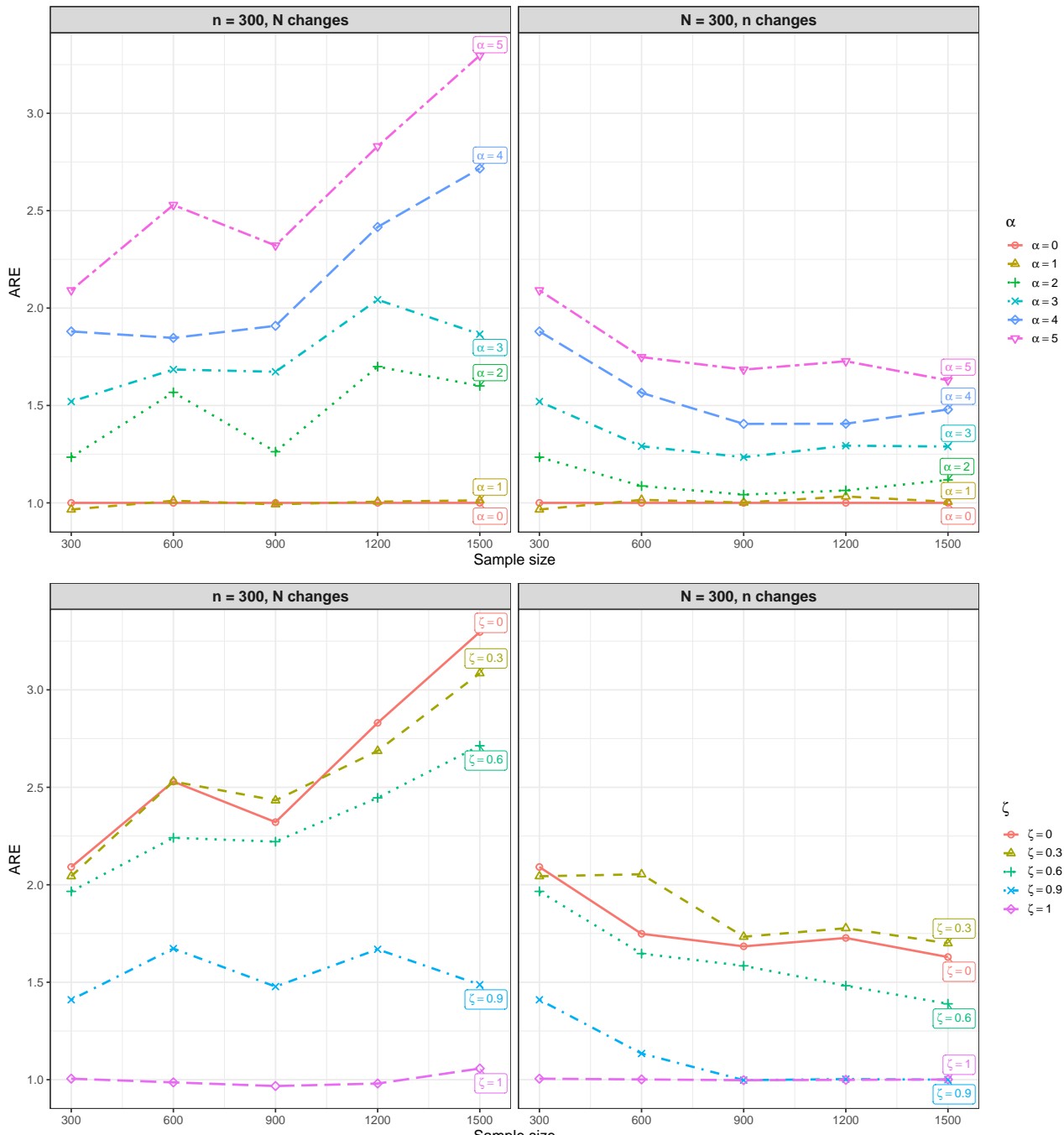

*Figure 8.* Variation of ARE for the estimation of $\xi_2$ under logistic model setting with different sample sizes, signal strength, and correlation coefficient.

| $N$ | Method | $\mathrm{MSE}(\overline{Y})$ | $\mathrm{MSE}(\widehat{\xi}_1)$ | $\mathrm{MSE}(\widehat{\xi}_2)$ | $\mathrm{MSE}(\widehat{\xi}_3)$ | $\mathrm{MSE}(\widehat{\xi}_4)$ | $\mathrm{MSE}(\widehat{\xi}_5)$ |
|---|---|---|---|---|---|---|---|
| 300 | w.o. ACP | 0.84 | 0.82 | 0.86 | 1.00 | 0.93 | 0.93 |
| 300 | w. ACP | 0.08 | 0.10 | 0.08 | 0.10 | 0.09 | 0.10 |
| 600 | w.o. ACP | 0.42 | 0.55 | 0.45 | 0.57 | 0.52 | 0.54 |
| 600 | w. ACP | 0.07 | 0.07 | 0.07 | 0.07 | 0.07 | 0.08 |
| 900 | w.o. ACP | 0.29 | 0.38 | 0.36 | 0.43 | 0.39 | 0.33 |
| 900 | w. ACP | 0.05 | 0.06 | 0.06 | 0.06 | 0.06 | 0.06 |
| 1200 | w.o. ACP | 0.24 | 0.30 | 0.29 | 0.32 | 0.31 | 0.25 |
| 1200 | w. ACP | 0.05 | 0.06 | 0.05 | 0.05 | 0.06 | 0.06 |
| 1500 | w.o. ACP | 0.21 | 0.26 | 0.27 | 0.29 | 0.27 | 0.25 |
| 1500 | w. ACP | 0.04 | 0.05 | 0.05 | 0.05 | 0.05 | 0.05 |

*Table 6.* Simulation results under different $N$ when $\alpha = 5$, $\zeta = 0$, $n = 300$ under the linear model setting.

| $\alpha$ | Method | $\mathrm{MSE}(\overline{Y})$ | $\mathrm{MSE}(\widehat{\xi}_1)$ | $\mathrm{MSE}(\widehat{\xi}_2)$ | $\mathrm{MSE}(\widehat{\xi}_3)$ | $\mathrm{MSE}(\widehat{\xi}_4)$ | $\mathrm{MSE}(\widehat{\xi}_5)$ |
|---|---|---|---|---|---|---|---|
| 0 | w.o. ACP | 0.03 | 0.03 | 0.03 | 0.04 | 0.03 | 0.03 |
| 0 | w. ACP | 0.03 | 0.03 | 0.03 | 0.04 | 0.03 | 0.03 |
| 1 | w.o. ACP | 0.06 | 0.07 | 0.06 | 0.07 | 0.07 | 0.07 |
| 1 | w. ACP | 0.03 | 0.03 | 0.03 | 0.04 | 0.04 | 0.03 |
| 2 | w.o. ACP | 0.16 | 0.16 | 0.15 | 0.18 | 0.17 | 0.17 |
| 2 | w. ACP | 0.04 | 0.04 | 0.04 | 0.05 | 0.04 | 0.04 |
| 3 | w.o. ACP | 0.32 | 0.32 | 0.32 | 0.37 | 0.35 | 0.34 |
| 3 | w. ACP | 0.05 | 0.06 | 0.05 | 0.06 | 0.05 | 0.06 |
| 4 | w.o. ACP | 0.54 | 0.54 | 0.56 | 0.65 | 0.60 | 0.59 |
| 4 | w. ACP | 0.06 | 0.07 | 0.06 | 0.08 | 0.07 | 0.08 |
| 5 | w.o. ACP | 0.84 | 0.82 | 0.86 | 1.00 | 0.93 | 0.93 |
| 5 | w. ACP | 0.08 | 0.10 | 0.08 | 0.10 | 0.09 | 0.10 |

*Table 7.* Simulation results under different $\alpha$ when $\zeta = 0$ and $n = N = 300$ under the linear model setting.

| $\zeta$ | Method | $\text{MSE}(\overline{Y})$ | $\text{MSE}(\widehat{\xi}_1)$ | $\text{MSE}(\widehat{\xi}_2)$ | $\text{MSE}(\widehat{\xi}_3)$ | $\text{MSE}(\widehat{\xi}_4)$ | $\text{MSE}(\widehat{\xi}_5)$ |
|---|---|---|---|---|---|---|---|
| 0 | w.o. ACP | 0.84 | 0.82 | 0.86 | 1.00 | 0.93 | 0.93 |
| 0 | w. ACP | 0.08 | 0.10 | 0.08 | 0.10 | 0.09 | 0.10 |
| 0.3 | w.o. ACP | 0.72 | 0.70 | 0.79 | 0.64 | 0.70 | 0.63 |
| 0.3 | w. ACP | 0.09 | 0.09 | 0.09 | 0.08 | 0.09 | 0.09 |
| 0.6 | w.o. ACP | 0.51 | 0.52 | 0.61 | 0.40 | 0.50 | 0.49 |
| 0.6 | w. ACP | 0.10 | 0.07 | 0.07 | 0.06 | 0.07 | 0.07 |
| 0.9 | w.o. ACP | 0.25 | 0.15 | 0.17 | 0.16 | 0.16 | 0.14 |
| 0.9 | w. ACP | 0.11 | 0.04 | 0.05 | 0.05 | 0.04 | 0.04 |
| 1 | w.o. ACP | 0.12 | 0.03 | 0.04 | 0.02 | 0.03 | 0.03 |
| 1 | w. ACP | 0.12 | 0.03 | 0.04 | 0.02 | 0.03 | 0.03 |

*Table 8.* Simulation results under different $\zeta$ when $\alpha = 5$ and $n = N = 300$ under the linear model setting.

| $n$ | Method | $\text{MSE}(\overline{Y})$ | $\text{MSE}(\widehat{\xi}_1)$ | $\text{MSE}(\widehat{\xi}_2)$ | $\text{MSE}(\widehat{\xi}_3)$ | $\text{MSE}(\widehat{\xi}_4)$ | $\text{MSE}(\widehat{\xi}_5)$ |
|---|---|---|---|---|---|---|---|
| 300 | w.o. ACP | 0.01 | 0.24 | 0.24 | 0.17 | 0.23 | 0.21 |
| 300 | w. ACP | 0.00 | 0.12 | 0.11 | 0.09 | 0.10 | 0.10 |
| 600 | w.o. ACP | 0.01 | 0.35 | 0.34 | 0.25 | 0.25 | 0.30 |
| 600 | w. ACP | 0.00 | 0.13 | 0.14 | 0.10 | 0.10 | 0.12 |
| 900 | w.o. ACP | 0.01 | 0.32 | 0.31 | 0.25 | 0.25 | 0.30 |
| 900 | w. ACP | 0.00 | 0.13 | 0.14 | 0.11 | 0.10 | 0.12 |
| 1200 | w.o. ACP | 0.01 | 0.39 | 0.44 | 0.28 | 0.28 | 0.37 |
| 1200 | w. ACP | 0.00 | 0.14 | 0.15 | 0.11 | 0.11 | 0.13 |
| 1500 | w.o. ACP | 0.01 | 0.41 | 0.48 | 0.30 | 0.28 | 0.35 |
| 1500 | w. ACP | 0.00 | 0.14 | 0.14 | 0.11 | 0.11 | 0.13 |

*Table 9.* Simulation results under different $n$ when $\alpha = 5$, $\zeta = 0$, $N = 300$ under the logistic model setting.

| $N$ | Method | $\text{MSE}(\overline{Y})$ | $\text{MSE}(\widehat{\xi}_1)$ | $\text{MSE}(\widehat{\xi}_2)$ | $\text{MSE}(\widehat{\xi}_3)$ | $\text{MSE}(\widehat{\xi}_4)$ | $\text{MSE}(\widehat{\xi}_5)$ |
|---|---|---|---|---|---|---|---|
| 300 | w.o. ACP | 0.01 | 0.24 | 0.24 | 0.17 | 0.23 | 0.21 |
| 300 | w. ACP | 0.00 | 0.12 | 0.11 | 0.09 | 0.10 | 0.10 |
| 600 | w.o. ACP | 0.00 | 0.10 | 0.11 | 0.08 | 0.10 | 0.09 |
| 600 | w. ACP | 0.00 | 0.06 | 0.06 | 0.05 | 0.06 | 0.06 |
| 900 | w.o. ACP | 0.00 | 0.07 | 0.07 | 0.06 | 0.07 | 0.06 |
| 900 | w. ACP | 0.00 | 0.04 | 0.04 | 0.03 | 0.04 | 0.04 |
| 1200 | w.o. ACP | 0.00 | 0.05 | 0.05 | 0.05 | 0.05 | 0.04 |
| 1200 | w. ACP | 0.00 | 0.03 | 0.03 | 0.03 | 0.03 | 0.03 |
| 1500 | w.o. ACP | 0.00 | 0.04 | 0.04 | 0.04 | 0.04 | 0.03 |
| 1500 | w. ACP | 0.00 | 0.02 | 0.02 | 0.02 | 0.02 | 0.02 |

*Table 10.* Simulation results under different $N$ when $\alpha = 5$, $\zeta = 0$, $n = 300$ under the logistic model setting.

| $\alpha$ | Method | MSE($\overline{Y}$) | MSE($\widehat{\xi}_1$) | MSE($\widehat{\xi}_2$) | MSE($\widehat{\xi}_3$) | MSE($\widehat{\xi}_4$) | MSE($\widehat{\xi}_5$) |
|---|---|---|---|---|---|---|---|
| 0 | w.o. ACP | 0.00 | 0.12 | 0.10 | 0.12 | 0.09 | 0.11 |
| 0 | w. ACP | 0.00 | 0.12 | 0.10 | 0.12 | 0.09 | 0.11 |
| 1 | w.o. ACP | 0.00 | 0.12 | 0.11 | 0.14 | 0.12 | 0.10 |
| 1 | w. ACP | 0.00 | 0.12 | 0.11 | 0.13 | 0.11 | 0.10 |
| 2 | w.o. ACP | 0.00 | 0.18 | 0.17 | 0.16 | 0.17 | 0.17 |
| 2 | w. ACP | 0.00 | 0.13 | 0.14 | 0.13 | 0.12 | 0.12 |
| 3 | w.o. ACP | 0.00 | 0.21 | 0.19 | 0.17 | 0.20 | 0.20 |
| 3 | w. ACP | 0.00 | 0.14 | 0.13 | 0.12 | 0.12 | 0.11 |
| 4 | w.o. ACP | 0.01 | 0.23 | 0.22 | 0.16 | 0.22 | 0.20 |
| 4 | w. ACP | 0.00 | 0.12 | 0.12 | 0.10 | 0.11 | 0.11 |
| 5 | w.o. ACP | 0.01 | 0.24 | 0.24 | 0.17 | 0.23 | 0.21 |
| 5 | w. ACP | 0.00 | 0.12 | 0.11 | 0.09 | 0.10 | 0.10 |

*Table 11.* Simulation results under different $\alpha$ when $\zeta = 0$ and $n = N = 300$ under the logistic model setting.

| $\zeta$ | Method | MSE($\overline{Y}$) | MSE($\widehat{\xi}_1$) | MSE($\widehat{\xi}_2$) | MSE($\widehat{\xi}_3$) | MSE($\widehat{\xi}_4$) | MSE($\widehat{\xi}_5$) |
|---|---|---|---|---|---|---|---|
| 0 | w.o. ACP | 0.01 | 0.24 | 0.24 | 0.17 | 0.23 | 0.21 |
| 0 | w. ACP | 0.00 | 0.12 | 0.11 | 0.09 | 0.10 | 0.10 |
| 0.3 | w.o. ACP | 0.00 | 0.25 | 0.23 | 0.19 | 0.19 | 0.22 |
| 0.3 | w. ACP | 0.00 | 0.11 | 0.11 | 0.09 | 0.09 | 0.10 |
| 0.6 | w.o. ACP | 0.00 | 0.25 | 0.21 | 0.19 | 0.22 | 0.20 |
| 0.6 | w. ACP | 0.00 | 0.13 | 0.10 | 0.10 | 0.10 | 0.09 |
| 0.9 | w.o. ACP | 0.00 | 0.70 | 0.15 | 0.13 | 0.14 | 0.12 |
| 0.9 | w. ACP | 0.00 | 0.65 | 0.11 | 0.12 | 0.13 | 0.11 |
| 1 | w.o. ACP | 0.00 | 7.38 | 0.15 | 0.17 | 0.17 | 0.16 |
| 1 | w. ACP | 0.00 | 7.11 | 0.15 | 0.16 | 0.17 | 0.16 |

*Table 12.* Simulation results under different $\zeta$ when $\alpha = 5$ and $n = N = 300$ under the logistic model setting.

*Table 13.* Comparison of mean squared error (MSE) for estimating **outcome mean** between proposed method and three alternatives: PPI, PPI++, RePPI, under the linear model setting. **Bolded values (smallest across all four methods)** indicate that the proposed method performs the best.

| Setting | Method | $N = 300$ | 600 | 900 | 1200 | 1500 |
|---|---|---|---|---|---|---|
| $\alpha = 5$ $\zeta = 0$ $n = 300$ | PPI | 2.8976 | 2.7768 | 2.6824 | 2.8998 | 2.9021 |
| | PPI++ | 2.8689 | 2.7499 | 2.6573 | 2.8496 | 2.8497 |
| | RePPI | 2.8639 | 2.7452 | 2.6528 | 3.8688 | 3.9909 |
| | proposed | **0.0803** | **0.0756** | **0.0712** | **0.0481** | **0.0451** |
| | Method | $n = 300$ | 600 | 900 | 1200 | 1500 |
| $\alpha = 5$ $\zeta = 0$ $N = 300$ | PPI | 2.8976 | 2.8811 | 2.8876 | 2.8897 | 2.9017 |
| | PPI++ | 2.8689 | 2.8711 | 2.8829 | 2.8899 | 2.9048 |
| | RePPI | 2.8639 | 0.9078 | 0.4444 | 0.2822 | 0.1898 |
| | proposed | **0.0803** | **0.0651** | **0.0545** | **0.0484** | **0.0441** |
| | Method | $\alpha = 0$ | 1 | 2 | 3 | 4 |
| $\zeta = 0$ $n = 300$ $N = 300$ | PPI | 2.8273 | 2.8364 | 2.8475 | 2.8588 | 2.8757 |
| | PPI++ | 2.8465 | 2.8470 | 2.8492 | 2.8524 | 2.8593 |
| | RePPI | 0.0583 | 0.1344 | 0.4725 | 1.0425 | 1.8403 |
| | proposed | **0.0327** | **0.0338** | **0.0393** | **0.0492** | **0.0630** |
| | Method | $\zeta = 0$ | 0.3 | 0.6 | 0.9 | 1 |
| $\alpha = 5$ $n = 300$ $N = 300$ | PPI | 2.8976 | 5.7063 | 9.3864 | 13.8583 | 15.6780 |
| | PPI++ | 2.8689 | 4.4729 | 6.2252 | 8.2971 | 9.0483 |
| | RePPI | 2.8639 | 2.2030 | 1.6693 | 0.5678 | 0.1476 |
| | proposed | **0.0803** | **0.0881** | **0.0990** | **0.1107** | **0.1231** |

*Table 14.* Comparison of mean squared error (MSE) for estimating **regression coefficients** between proposed method and three alternatives: PPI, PPI++, RePPI, under the linear model setting. **Bolded values (smallest across all four methods)** indicate that the proposed method performs the best.

| $N$ | Method | $X_1$ | $X_2$ | $X_3$ | $X_4$ | $X_5$ |
|---|---|---|---|---|---|---|
| 300 | PPI | 0.1290 | 0.1136 | 0.1233 | 0.1158 | 0.1140 |
| | PPI++ | 0.1086 | 0.1086 | 0.1178 | 0.0987 | 0.1033 |
| | RePPI | 0.2463 | 0.2741 | 0.2713 | 0.2676 | 0.2642 |
| | propoesd | **0.0975** | **0.0844** | **0.0977** | **0.0889** | **0.0989** |
| 600 | PPI | 0.1208 | 0.1072 | 0.1156 | 0.1102 | 0.1091 |
| | PPI++ | 0.1035 | 0.1029 | 0.1113 | 0.0962 | 0.0994 |
| | RePPI | 0.2381 | 0.2613 | 0.2587 | 0.2542 | 0.2506 |
| | proposed | **0.0913** | **0.0792** | **0.0934** | **0.0851** | **0.0948** |
| 900 | PPI | 0.1137 | 0.1015 | 0.1092 | 0.1045 | 0.1033 |
| | PPI++ | 0.0992 | 0.0985 | 0.1067 | 0.0923 | 0.0951 |
| | RePPI | 0.2297 | 0.2498 | 0.2471 | 0.2436 | 0.2401 |
| | proposed | **0.0867** | **0.0751** | **0.0889** | **0.0813** | **0.0912** |
| 1200 | PPI | 0.1219 | 0.1044 | 0.1166 | 0.1090 | 0.1084 |
| | PPI++ | 0.1219 | 0.1044 | 0.1166 | 0.1090 | 0.1084 |
| | RePPI | 0.3001 | 0.3396 | 0.3337 | 0.3255 | 0.3133 |
| | proposed | **0.0493** | **0.0434** | **0.0548** | **0.0481** | **0.0483** |
| 1500 | PPI | 0.1204 | 0.1038 | 0.1158 | 0.1080 | 0.1069 |
| | PPI++ | 0.1204 | 0.1038 | 0.1158 | 0.1080 | 0.1069 |
| | RePPI | 0.3088 | 0.3465 | 0.3444 | 0.3370 | 0.3170 |
| | proposed | **0.0431** | **0.0408** | **0.0486** | **0.0426** | **0.0417** |

