# OpenReview forum: "Towards the Efficient Inference by Incorporating Automated Computational Phenotypes under Covariate Shift"
_ICML.cc/2025/Conference — ICML 2025 poster_

### Official Review · Reviewer_pTuV · 2025-03-15

**Overall Recommendation:** 3

**Summary:**

This paper explores the integration of automated computational phenotypes (ACPs) in semi-supervised learning settings. ACPs are used to derive phenotype data from electronic health records (EHRs) using machine learning models, reducing the labor-intensive nature of manual phenotype extraction. However, direct replacement of gold-standard phenotype data with ACPs can introduce bias.

**Claims And Evidence:**

The paper provides rigorous mathematical justifications and asymptotic efficiency analyses to show the advantages of using ACPs.
The empirical results from simulations and a real-world application to diabetes prediction support the claim that ACPs improve inference efficiency. The paper assumes that the ACP prediction errors do not introduce systemic biases beyond covariate shift, which might not always hold in real-world scenarios. A sensitivity analysis could be conducted to investigate this.
However, the practical effectiveness of these methods in more complex, real-world medical settings (beyond a single dataset) could be better validated.

**Essential References Not Discussed:**

The authors can perhaps also discuss the relationship with the paper Doubly Robust Calibration of Prediction Sets under Covariate shift (Yang, Kuchibhotla, Tchetgen Tchetgen 2024), which is highly relevant to their work.

**Experimental Designs Or Analyses:**

No issue, would be better to have more experiments.

**Methods And Evaluation Criteria:**

The paper defines clear evaluation metrics (asymptotic efficiency bounds, mean squared error comparisons).
They compare scenarios with and without ACPs to show how the additional data impacts estimation.
The use of cross-fitting for nuisance parameter estimation ensures robustness when applying machine learning methods.
 It does not extensively discuss computational trade-offs (e.g., increased complexity due to doubly robust estimation).

**Other Comments Or Suggestions:**

None

**Other Strengths And Weaknesses:**

The efficiency analysis is mathematically well-founded and there is a experiment on both real world and synthetic datasets.
Some of the assumptions are too strong.

**Questions For Authors:**

NA

**Relation To Broader Scientific Literature:**

The paper is well-grounded in existing literature on: Semi-supervised learning (e.g., Zhu 2005, Rigollet 2006, Wang et al. 2022).
Inference with predicted data (prediction-powered inference, PPI) (Angelopoulos et al. 2023).
Covariate shift and domain adaptation (Sugiyama et al. 2008, Gretton et al. 2009). Surrogacy in biostatistics and causal inference (Athey et al. 2019, Imbens et al. 2024).
The paper’s main contribution is combining semi-supervised learning, covariate shift adjustments, and ACPs in a unified framework. The paper does not extensively compare its approach to existing robust semi-supervised estimation methods beyond theoretical efficiency arguments. T

**Theoretical Claims:**

I didn't see any issue.
Some technical assumptions (e.g., regularity conditions on nuisance estimators, independence assumptions for ACP generation) are not explicitly tested.

---

> ### Author Rebuttal · Authors · 2025-04-01
>
> *Q1: [Claims and Evidence]*
>
> **Response:**
> - Thank you for this insightful comment! To understand the benefits of incorporating ACP $\hat Y$, we introduce an assumption with a format similar to covariate shift, as presented in Lines 181-192. In real world scenarios, this assumption can be tested when $Y$ is available in the unlabeled data (e.g., in the income, politeness, wine datasets) but not when $Y$ is unavailable (e.g., in the diabetes dataset). In cases where the assumption cannot be tested, we agree that conducting a sensitivity analysis could be valuable and of help.
> - We also agree that evaluations on more datasets are worthwhile. We provided a more comprehensive response in our answer to Q2 of reviewer mNmv and to Q4 of reviewer 6g7c. Please kindly refer to that section for details. Briefly we newly analyze three datasets. For each of them, we present the comparison results of confidence interval length for regression coefficients in Table 1 of [More Results](https://drive.google.com/file/d/1ihn6OAqb26TZ0KVaelSpWO1bQzqGDQaH/view?usp=sharing). The proposed method is generally a lot more efficient than PPI, PPI++, and RePPI, with the largest reduction reaching approximately 60%.
>
> *Q2: [Methods and Evaluation Criteria]*
>
> **Response:**
> - This is definitely an excellent point that warrants further discussion in our revised version. In the presence of covariate shift, one must account for two sources of nuisance functions: the density ratio model and the regression model. Estimating both sources of nuisance functions is computationally more complex than estimating just one, as in some alternative methods. However, the inclusion of both sources forms the basis of double robustness, which is essential for achieving efficient estimation. This represents a tradeoff between estimation efficiency and computational complexity. In cases where the mechanism for selecting labels is known, such as in certain design-based studies, the proposed method only needs to estimate one source of nuisance function.
>
> *Q3: [Theoretical Claims]*
>
> **Response:**
> - The regularity conditions on nuisance estimators are standard and widely advocated in double/debiased ML literature, e.g., van der Laan (2011), Chernozhukov et al. (2018), Kennedy (2024), Chernozhukov et al. (2024). The convergence rate is achievable for many ML methods such as in regression trees and random forests (Wager and Walther 2015) and a class of neural nets (Chen and White 1999). One can refer to hernozhukov et al. (2018) for more examples.
> - As we briefly mentioned in our answer to Q1, the independence assumption for ACP generation can be tested when $Y$ is available in the unlabeled data but cannot if otherwise. When it cannot be tested, one can decompose the assumption as $p(\hat y|x) = q(\hat y|x)$ and $p(y|x, \hat y) = q(y|x,\hat y)$. It is clear that, the untestable part $p(y|x, \hat y)=q(y|x,\hat y)$ is something similar to the covariate assumption $p(y|x) = q(y|x)$. Note that this assumption is popularly adopted in semi-supervised learning, and distribution shift.
>
> *Q4: [Experimental Designs or Analyses]*
>
> **Response:**
> - We implemented our proposed method and compared with some alternatives on three new datasets (income, politeness, wine). We provided a more comprehensive response in our answer to Q2 of reviewer mNmv, to Q4 of reviewer 6g7c, and to Q1 above. Briefly, we present the comparison results of confidence interval length for regression coefficients in Table 1 of [More Results](https://drive.google.com/file/d/1ihn6OAqb26TZ0KVaelSpWO1bQzqGDQaH/view?usp=sharing). The proposed method is generally a lot more efficient than PPI, PPI++, and RePPI, with the largest reduction reaching approximately 60%.
> - We also expanded our simulation studies by comparing the proposed method with PPI, PPI++ and RePPI. We provided a more comprehensive response in our answer to Q2 of reviewer 6g7c. Briefly, the MSE results are presented in Tables 2 and 3 of [More Results](https://drive.google.com/file/d/1ihn6OAqb26TZ0KVaelSpWO1bQzqGDQaH/view?usp=sharing). Across all considered scenarios, the proposed method consistently outperforms the three alternatives.
>
> *Q5: [Relation to Broader Scientific Literature]*
>
> **Response:**
> - As in our answers to Q1 and Q4 above, we extensively compare the proposed method with PPI, PPI++ and RePPI in both real data and simulated data.
>
> *Q6: [Essential References Not Discussed]*
>
> **Response:**
> - Thank you for pointing out this important reference! Indeed, Yang et al. (2024) studies how to calibrate prediction sets in the presence of covariate shift and proposes a doubly robust approach to enhance the reliability and coverage of predictions. We also realized that there is work, such as Qiu et al. (2023), that explores prediction sets adaptable to unknown covariate shift. In the revised version, we will be sure to include these relevant, important, and exciting works!
>
> *Q7: [Other Strengths And Weaknesses]*
>
> **Response:**
> - Same as Q3 above.

---

> > ### Comment · Reviewer_pTuV · 2025-04-02
> >
> > The additional experiments are satisfactory, I would like to change my score to 4.

---

> > > ### Author Response · Authors · 2025-04-02
> > >
> > > Dear Reviewer pTuV,
> > >
> > > Thank you once again for your thoughtful review of our work, your insightful comments, and for increasing your score from 3 to 4, as indicated in your response to the rebuttals.
> > >
> > > We noticed, however, that the submission summary page still reflects your original score of 3. We’re wondering if the score change might not have been updated in the system.
> > >
> > > Could you kindly take a moment to check and ensure the updated score is reflected? We truly appreciate your time and support.
> > >
> > > Sincerely,
> > > The Authors

---

### Official Review · Reviewer_6g7C · 2025-03-17

**Overall Recommendation:** 4

**Summary:**

This paper introduces a semiparametric framework for efficient inference under covariate shift by leveraging automated computational phenotypes (ACPs). The authors propose a doubly robust, semiparametrically efficient estimator for a target parameter $\beta$ by integrating ACPs, density ratios, and conditional expectations.

Theoretical results show that when ACPs offer extra predictive information beyond $\boldsymbol{X}$, their inclusion strictly reduces the estimator's asymptotic variance. Simulation studies and a diabetes case study empirically validate that the proposed method achieves lower mean squared error and yields narrower confidence intervals compared to methods without ACPs.

**Claims And Evidence:**

Yes. Overall, the paper’s claims are supported by theoretical derivations and experiments.

**Essential References Not Discussed:**

[1] proposes an approach for semi-supervised learning algorithms that can address covariate shift. Their framework also recovers some popular methods, including entropy minimization and pseudo-labeling.
[2] discussed the use of pseudo-labeled data.

[1] Aminian, Gholamali, et al. "An information-theoretical approach to semi-supervised learning under covariate-shift." International Conference on Artificial Intelligence and Statistics. PMLR, 2022.


[2] Rodemann, Julian, et al. "In all likelihoods: Robust selection of pseudo-labeled data." International Symposium on Imprecise Probability: Theories and Applications. PMLR, 2023.

**Experimental Designs Or Analyses:**

1. The simulations compare non-ACP and using-ACP settings. While this is useful for demonstrating efficiency gains, a comparison with alternative methods from the literature (e.g., Prediction-Powered Inference) might further validate the practical benefits of the approach.
2. Why SuperLearner is used? Are there other options?
3. For real dataset, additional different datasets or disease contexts would strengthen the generalizability of the results.
4. For the diabetes dataset, how well does the covariate shift assumption hold?
5. For the diabetes dataset,  what $\boldsymbol{X}$, $\boldsymbol{Y}$, and $\boldsymbol{Z}$ mean?

**Methods And Evaluation Criteria:**

Yes. I think the use of ACP and covariate shift problem should be a typical problem. And the experiment on real dataset looks like a reasonable setting.
However, the author should have indicated what $\boldsymbol{X}$, $\boldsymbol{Y}$, and $\boldsymbol{Z}$ mean in those experiments, especially for the diabetes experiment.

**Other Comments Or Suggestions:**

NA

**Other Strengths And Weaknesses:**

NA

**Questions For Authors:**

In line 423-425, "We adopted a logistic regression to predict diabetes status with all selected variables,
and use the regression coefficients as association measures"

What linear/logistic regression is used? Did the authors try different models?
What kind of models can be analyzed with the proposed theory? For example, can large deep learning models be explored with this theory? What if MLP-based regressors or classifiers?
Can $\boldsymbol{X}$ be images or other modalities?

**Relation To Broader Scientific Literature:**

The paper’s contributions are related to established statistical theories such as semiparametric theory, double robustness, and semi-supervised learning under covariate shift. This paper focuses specifically on the use of ACP.
This paper also relates to some work about data imputation, the use of pseudo-labeled data, and epistemic uncertainty.

**Theoretical Claims:**

I reviewed the theoretical claims provided in the paper.
I did not perform a line-by-line check of every technical detail.
I think the proofs for the main theoretical claims appear correct.

---

> ### Author Rebuttal · Authors · 2025-04-01
>
> *Q1: [Methods and Evaluation Criteria]*
>
> **Response:**
> - Thank you. We provided a more comprehensive response in our answer to Q1 of reviewer mNmv. Please kindly refer to that section for details.
>
> *Q2: The simulations...[Experimental Designs or Analyses]*
>
> **Response:**
> - Yes. We conducted comparisons between the proposed method and PPI, PPI++, RePPI, across a variety of settings with different values of $n$, $N$, $\alpha$ and $\zeta$. The MSE results are presented in Tables 2 and 3 of [More Results](https://drive.google.com/file/d/1ihn6OAqb26TZ0KVaelSpWO1bQzqGDQaH/view?usp=sharing). As you can see, across all considered scenarios, the proposed method consistently outperforms the three alternatives.
>
> *Q3: Why SuperLearner...*
>
> **Response:**
> - We use Super Learner in our implementation, which integrates a library of flexible statistical learning tools to ensure consistent estimation of relevant nuisance functions. While these functions can be estimated using simple parametric models, such models are prone to misspecification, potentially introducing bias and reducing efficiency. More flexible machine learning methods mitigate this issue by avoiding strict parametric assumptions and enabling consistent estimation across a broader range of function classes. Instead of relying on a single estimation method, Super Learner leverages the strengths of multiple algorithms by constructing an optimally weighted combination, offering greater robustness and flexibility. Moreover, van der Laan et al. (2007) established a theoretical guarantee that the estimation error of Super Learner converges to that of the best-performing learner in the ensemble.
> - Alternatively, we may use individual learning algorithms, such as random forests or XGBoost. Neural networks are also a potential option, though the size of labeled data in our dataset---such as the diabetes data---is relatively small compared to typical deep learning applications.
>
> *Q4: For real dataset...*
>
> **Response:**
> - Yes we completely agree with you. We provided a more comprehensive response in our answer to Q2 of reviewer mNmv. Please kindly refer to that section for details. Briefly we newly analyze three datasets. For each of them, we present the comparison results of confidence interval length for regression coefficients in Table 1 of [More Results](https://drive.google.com/file/d/1ihn6OAqb26TZ0KVaelSpWO1bQzqGDQaH/view?usp=sharing). The proposed method is generally a lot more efficient than PPI, PPI++, and RePPI, with the largest reduction reaching approximately 60\%.
>
> *Q5: For the diabetes dataset...*
>
> **Response:**
> - Figure 2 in our submitted paper demonstrates that the distribution of inpatient visit count indeed shifts between labeled and unlabeled data. To provide a more comprehensive comparison, Table 4 of [More Results](https://drive.google.com/file/d/1ihn6OAqb26TZ0KVaelSpWO1bQzqGDQaH/view?usp=sharing) presents the summary statistics of each variable in $X$. Overall, the covariate shift assumption appears reasonable in this dataset.
>
> *Q6: For the diabetes dataset...*
>
> **Response:**
> - Same as Q1 above.
>
> *Q7: [Supplementary Material]*
>
> **Response:**
> - Thank you for the suggestion, and we apologize for overlooking this issue. In the new version, we will make sure to appropriately reference the corresponding details in the supplementary material within the main paper.
>
> *Q8: [Essential References Not Discussed]*
>
> **Response:**
> - Thank you for pointing out these references! Both are highly exciting works. In the new version, we will be sure to include them and conduct a more thorough literature review, particularly on the topics of covariate shift and pseudo-labeling.
>
> *Q9: [Questions for Authors]*
>
> **Response:**
> - In the diabetes dataset, the scientific goal is to understand the relationship between $Y$ (diabetes status) and $X$ (7 variables representing patient characteristics). Since $Y$ is binary, we use logistic regression, which is arguably the most commonly used model for this type of analysis. More generally, other suitable regression or classification models, such as probit regression or support vector machines, could also be applied in this context.
> - In a broader sense, as long as the parameter of interest can be defined as the minimizer of a loss function, as presented in Lines 144-146, the proposed method can be used. This includes the case with MLP-based regressors or classifiers, high-dimensional models, or large deep learning models in general. To address this question more explicitly, we explore the implementation of the proposed method on the benchmark dataset MNIST where $X$ is image, $\hat Y$ is generated using ResNet, and the comparison with the standard PPI method, for estimating the probability of the outcome being a certain label. In terms of the length of 95\% confidence interval (the shorter, the more efficiency gain), the proposed method shortens about 15\% (from 0.0448 to 0.0376 when $Y$=1/7, and from 0.0440 to 0.0370 when $Y$=6/8).

---

> > ### Comment · Reviewer_6g7C · 2025-04-03
> >
> > Thanks for the rebuttal. My questions are solved. I appreciate the additional experiments and illustrations about the dataset and experiment setting. I have increased my score to 4.

---

> > > ### Author Response · Authors · 2025-04-03
> > >
> > > Dear Reviewer 6g7c,
> > >
> > > Thank you so much for your acknowledgement, your thoughtful reviews of our work, and for increasing the score! We really appreciate it!
> > >
> > > Sincerely
> > > The Authors

---

### Official Review · Reviewer_mNmv · 2025-03-18

**Overall Recommendation:** 3

**Summary:**

This paper proposes an approach that leverages both labeled and unlabeled data to estimate target parameters. The approach first uses the pre-trained model to estimate $Y$ for the unlabeled data and then uses the estimated $\hat{Y}$ to estimate the target parameters. The proposed approach is applied to both simulated and real-world data to demonstrate the efficiency gain as compared to the benchmark approach.

**Claims And Evidence:**

The proposed approach is supported by theoretical analysis and empirical study.

**Essential References Not Discussed:**

NA

**Experimental Designs Or Analyses:**

The experimental results make sense. However, I would appreciate more real-world experiments (on other data sets) to demonstrate that the proposed approach has a substantial efficiency gain.

**Methods And Evaluation Criteria:**

It makes intuitive sense to augment the labeled data with unlabeled data to estimate target parameters. The paper shows in Section 3.2 that the unlabeled data can improve the estimation efficiency of target parameters. However, it is a bit unclear when the efficiency gain can actually happen, i.e., when the generation of $Y$ depends on covariates $X$ and other variables $Z$. Could the authors provide a concrete example? What are $X$, $Z$, and $Y$? When $Z$ is used to generate $Y$ but is not included in $X$ itself?

**Other Comments Or Suggestions:**

Please address my comments on methods and experiments.

**Other Strengths And Weaknesses:**

NA

**Questions For Authors:**

NA

**Relation To Broader Scientific Literature:**

This paper is related to the literature on semi-supervised learning and prediction-powered inference.

**Theoretical Claims:**

The theoretical claims look correct to me.

---

> ### Author Rebuttal · Authors · 2025-04-01
>
> *Q1: However, it is a bit unclear...[from Methods and Evaluation Criteria]*
>
> **Response**:
> - Thanks for raising this question. In general, the outcome $Y$ and covariate $X$ are variables of scientific interest. The variable $Z$ represents additional information that may not be of direct scientific interest but is highly relevant to $Y$.
> - For example, in our diabetes dataset, $Y$ indicates whether a patient has diabetes, and $X$ includes 7 variables measured at the year of diagnosis: age, type of insurance (self-pay vs others), counts of inpatient and outpatients visits, BMI, congestive heart failure (yes vs no), and the Charlson Comorbidity Index (CCI, larger than 2 or not). Our primary scientific interest is in understanding the relationship between $Y$ and $X$, which captures patient characteristics. Meanwhile, ACP $\hat Y$ is generated by a tree-based algorithm using diagnosis codes, medication history, and HbA1c lab test results from electronic health records---these variables form $Z$.
> - In another example using newly analyzed income data, we fit a least squares model to study the relationship between $Y$ (log-income) and $X$ (age, sex). The ACP $\hat Y$ is obtained from an XGBoost model predicting log-income based on 14 variables, including education, marital status, citizenship, and race, among others---these variables consititute $Z$.
> - In some cases where $\hat Y$ is generated by LLMs, defining $Z$ precisely is more challenging. For instance, in our politeness dataset, the goal is to study the relationship between $Y$ (politeness score) and $X$ (a binary indicator of hedging in requests), while $\hat Y$ is generated by OpenAI's GPT-4o mini model. Here, $Z$ represents some latent information used to produce $\hat Y$ but is not as explicitly defined as in previous examples. Similarly, in our wine dataset, the goal is to examine the relationship between $Y$ (rating) and $X$ (price and region), with $\hat Y$ also generated by GPT-4o mini.
> - Finally, in a synthetic data example, consider the following data generating process $Y = \xi^T X + \alpha Z + \varepsilon$, where $\varepsilon \sim N(0,1)$, $\alpha>0$, as in our simulation studies. Suppose the parameter of interest is $\xi$, and $\hat Y = Z$. The magnitude of $\alpha$ determines whether efficiency gain occurs: as long as $\alpha\neq 0$, $Z$ correlates with $Y$, leading to an efficiency gain.
>
> *Q2: However, I would appreciate...[from Experimental Designs or Analyses]*
>
> **Response**:
> - Yes. Thank you for this question. We implemented the proposed method on three new data sets, and compared with some existing methods: PPI (Angelopoulos et al. 2023a), PPI++ (Angelopoulos et al. 2023b) and RePPI (Ji et al. 2025).
> - Income data: We analyze the relationship between wage (measured by log-income) and age, confounded by sex, based on US census data, under covariate shift. The ACP $\hat Y$ is generated by fitting XGBoost of log-income with 14 variables, including education, marital status, citizenship, and race. For covariate shift, we partition the labeled and unlabeled datasets using the probability $\exp(\alpha^T X)/\{1+\exp(\alpha^T X)\}$ where $\alpha=(0,1,0)$ and $X =(1,X_1,X_2)$ with $X_1$ age and $X_2$ sex, resulting in a ratio of 2:8.
> - Politeness data: Using data that comprises texts from 5,512 online requests posted on Stack Exchange and Wikipedia, we understand the association between politeness score (range from 1 to 25) and a binary indicator for hedging within the request. The ACP $\hat Y$ is generated using OpenAI's GPT-4o mini-model that has the same range as the politeness score. For covariate shift, we split the labeled and unlabeled data in a 1:9 ratio, following the same procedure as above, where $X=(1,X_1)$ with $X_1$ hedge and $\alpha=(0,1)$.
> - Wine data: Using the Wine Enthusiast review dataset, we investigate the association between wine rating (range from 80 to 100) and wine price, adjusted by wine region. Similar to the politeness data, the ACP $\hat Y$ is also generated by employing OpenAI's GPT-4o mini-model that produces predicted ratings with the same scale. To assess covariate shift, we follow the same procedure as in the previous experiments, splitting the labeled and unlabeled data in a 3:7 ratio. Here, $X=(1,X_1,X_2,X_3,X_4,X_5)$, where $X_1$ represents price, $X_2$ to $X_5$ represents California, Washington, Oregon and New York, respectively, with $\alpha=(0,1,0,0,0,0)$.
> - For each of these three data sets, we present the comparison results of confidence interval length (the shorter, the more efficient, the better) for regression coefficients in Table 1 of [More Results](https://drive.google.com/file/d/1ihn6OAqb26TZ0KVaelSpWO1bQzqGDQaH/view?usp=sharing).
> As you can see, while there are some cases that the proposed method is tiny slightly less efficient than PPI++ or RePPI, it is generally a lot more efficient than PPI, PPI++, and RePPI, with the largest reduction in confidence interval length reaching approximately 60%.

---

### Decision · Program_Chairs · 2025-05-01

**Decision:**

Accept (poster)

**Comment:**

The paper analyzes the efficiency of incorporating automated computational phenotypes (ACPs) in a semi-supervised learning setting, particularly in the case where a covariate shift exists between labeled and unlabeled data. For the estimation of a target parameter of the unlabeled(or combined) population, the authors derive the efficient influence function and the semiparametric efficiency bound under various scenarios, and rigorously show that ACPs can help improve efficiency when they are provided for the unlabeled data, and as long as they do not depend solely on available features. Finally, authors also propose a semi-parametrically efficient and doubly-robust estimator which also permits the use of flexible ML methods for estimation of nuisance functions.

Reviewers acknowledge the theoretical contribution of the paper, and the additional experiments submitted during the rebuttal further support the theoretical claims.